# Impact of tidal dynamics on diel vertical migration of zooplankton in Hudson Bay

Vladislav Y. Petrusevich[1*], Igor A. Dmitrenko[1], Andrea Niemi[2], Sergey A. Kirillov[1], Christina Michelle Kamula[1], Zou Zou A. Kuzyk[1], David G. Barber[1] and Jens K. Ehn[1]

[1]University of Manitoba, Centre for Earth Observation Science, Winnipeg, Canada
[2]Fisheries and Oceans Canada, Winnipeg, Manitoba, Canada

*Correspondence to*: Vladislav Y. Petrusevich (vlad.petrusevich@umanitoba.ca)

**Abstract.** Hudson Bay is a large, seasonally-ice covered Canadian inland sea, connected to the Arctic Ocean and North Atlantic through Foxe Basin and Hudson Strait. This study investigates zooplankton distribution, dynamics and factors controlling them during open water and ice cover periods (from September 2016 to October 2017) in Hudson Bay. A mooring equipped with two Acoustic Doppler Current Profilers (ADCP) and a sediment trap was deployed in September 2016 in Hudson Bay ~190 km north-east from the port of Churchill. The backscatter intensity and vertical velocity time series showed a pattern typical for the zooplankton diel vertical migration (DVM). The sediment trap collected five zooplankton taxa including two calanoid copepods (*Calanus glacialis* and *Pseudocalanus* spp.), a pelagic sea snail (*Limacina helicina*), a gelatinous arrow worm (*Parasagitta elegans*) and an amphipod (*Themisto libellula*). From the acquired acoustic data we observed the interaction of DVM with multiple factors including lunar light, tides, as well as water and sea ice dynamics. Solar illuminance was the major factor determining migration pattern, but unlike at some other polar and sub-polar regions, moonlight had a little effect on DVM, while tidal dynamics is important. The presented data constitutes a first-ever observed presence of DVM in Hudson Bay during winter as well as its interaction with the tidal dynamics.

## 1. Introduction

The diel vertical migration (DVM) of zooplankton is a synchronized movement of individuals through the water column and is considered to be the largest daily synchronized migration of biomass in the ocean (Brierley, 2014). This migration is majorly controlled by two biological factors: (1) predator avoidance by staying away from the illuminated surface layer during the day and thus reducing the light-dependent mortality risk (Hays, 2003; Ringelberg, 2010; Torgersen, 2003) and (2) optimization of feeding, with the assumption that algal biomass is greater in the surface layer during evening hours and zooplankton rise to feed on it in the evening (Lampert, 1989). There are three general DVM patterns: (1) The most common one is *nocturnal* when zooplankton ascends around sunset and remains at upper depth during the night, around sunrise descending and remaining at depth during the day (Cisewski et al., 2010; Cohen and Forward, 2002). (2) Then there is a *reverse* pattern when zooplankton is ascending up at dawn and descending at dusk (Heywood, 1996; Pascual et al., 2017). And finally, (3) there is *twilight* DVM pattern when zooplankton is ascending at sunset, then descending around midnight, then again ascending and finally descending at sunset (Cohen and Forward, 2005; Valle-Levinson et al., 2014). This pattern sometimes is called *midnight sink*.

DVM of zooplankton is an important process of the carbon and nitrogen cycle in marine systems, because it effectively acts as a biological pump, transporting carbon and nitrogen vertically below the mixed layer by respiration and excretion (Darnis et al., 2017; Doney and Steinberg, 2013; Falk-Petersen et al., 2008). The following research question needs to be addressed:

what sets the timing of this synchronized movement in the Arctic environment?Earlier studies of DVM in the Arctic were focused on the period of midnight sun or the transition period from midnight sun to a day/night cycle (Blachowiak-Samolyk et al., 2006; Cottier et al., 2006; Falk-Petersen et al., 2008; Fortier et al., 2001; Kosobokova, 1978; Rabindranath et al., 2010). Recent studies based on acoustic backscatter data and zooplankton sampling showed the presence of synchronized DVM behaviour continuing throughout the Arctic winter, during both open and ice-covered waters (Båtnes et al., 2015; Benoit et al.,

2010; Berge et al., 2009, 2012, 2015a, 2015b; Cohen et al., 2015; Last et al., 2016; Petrusevich et al., 2016; Wallace et al., 2010). It was proposed (Berge et al., 2014; Hobbs et al., 2014; Last et al., 2016; Petrusevich et al., 2016) that, during polar night, DVM is regulated by diel variations in solar and lunar illumination, which are at intensities far below the threshold of human perception. Another reason for increasing interest in studying DVM patterns in various geophysical and geographical environments and their seasonal changes in response to changing oceanographic conditions is that they could help inform us

about physical oceanographic processes. Furthermore, DVM pattern can be significantly modified by water column stratification (Berge et al., 2014) and water dynamics, such as polynya induced estuarine-like circulation (Petrusevich et al., 2016), tidal currents (Hill, 1991, 1994; Valle-Levinson et al., 2014), and upwelling/downwelling (Dmitrenko et al., 2019; Wang et al., 2015).

In the Arctic Ocean, the DVM process can be difficult to measure. However, there has been recent success in using data

obtained by an Acoustic Doppler Current Profiler (ADCP), which is a modern oceanographic instrument commonly used to measure the vertical profile of current velocities. Because the velocity profiling by an ADCP is based on processing the measured intensity of acoustic pings backscattered by suspended particles in the water column, further processing of the measured acoustic backscatter to volume backscatter strength (Deines, 1999) has been successful in quantifying zooplankton abundance (Bozzano et al., 2014; Brierley et al., 2006; Cisewski et al., 2010; Cisewski and Strass, 2016; Fielding et al., 2004;

Guerra et al., 2019; Hobbs et al., 2018; Last et al., 2016; Lemon et al., 2008; Petrusevich et al., 2016; Potiris et al., 2018, etc.). ADCP backscatter data, validated using a time-series of zooplankton samples collected from sediment traps, provides a particularly useful tool for understanding the effects of physical oceanographic processes on zooplankton DVM, changes in zooplankton community composition throughout the year, and an overall better understanding of the marine ecosystem function and carbon cycling (Berge et al., 2009; Willis et al., 2006, 2008).

In this study, we are focused on zooplankton organisms with sizes from 500 μm and up. This group of zooplankton primarily detected by ADCP backscatter (Cisewski and Strass, 2016; Pinot and Jansá, 2001) and allows comparison with previous studies on zooplankton caught by sediment traps (see Forbes et al., 1992; Pospelova et al., 2010).

In this study, factors controlling zooplankton distribution during the open-water and ice-covered periods are investigated using ADCP data together with sediment trap samples for the first time in Hudson Bay. The main objectives are to (1) examine DVM

during open water and ice-covered seasons in Hudson Bay in 2016-2017, (2) identify zooplankton species involved in DVM and (3) describe the DVM response to solar and lunar light, tides, water and sea-ice dynamics.

## 2. Study Area

Hudson Bay (Figure 1a) is a large (with an area about 831,000 km$^2$) seasonally ice-covered shallow inland sea with an average depth of 125 m and maximum depth below 300 m (Burt et al., 2016; Ingram and Prinseberg, 1998; Macdonald and Kuzyk, 2011; Petrusevich et al., 2018; St-Laurent et al., 2008; Straneo and Saucier, 2008). The seabed is characterized by fluted tills, postglacial infills, moraines and subglacial channels eroded to bedrock resulting in bottom depth varying from two hundred meters to ~10 m (Josenhans and Zevenhuizen, 1990). The tides are mostly lunar semidiurnal ($M_2$) with an amplitude of about 3m at the entrance to Hudson Bay from Hudson Strait (Prinsenberg and Freeman, 1986; St-Laurent et al., 2008) and about 1.5m in Churchill (Prinsenberg, 1987; Saucier et al., 2004)l (Figure 1) (Ray, 2016). The marine water masses flow into Hudson Bay through two gateways: (1) Gulf of Boothia – Fury and Hecla Strait – Foxe Basin, and (2) the Baffin Bay – Hudson Strait (Fig. 1a). Measurements of alkalinity and nutrient ratios suggest that the water masses within Hudson Bay are dominated by Pacific-origin waters from the Arctic Ocean (Burt et al., 2016; Jones et al., 2003) and the phytoplankton and zooplankton assemblages resemble those in the Arctic Ocean (Estrada et al., 2012; Runge and Ingram, 1991). Freshwater inputs to Hudson Bay are very large, including river runoff from the largest watershed in Canada, together with seasonal inputs of sea ice-melt. The freshwater inputs together produce strong stratification at the surface in summer (Ferland et al., 2011). Fall storms and cooling followed by brine rejection from sea ice formation during winter produces a winter surface mixed layer varying from ~40 to >90 m deep throughout Hudson Bay (Prinsenberg, 1987; Saucier et al. 2004).

Hudson Bay is ice-covered during 7–9 months a year with ice formation typically starting in the north-west part of the bay in late October (Hochheim and Barber, 2014). The mean maximum ice thickness ranges from 1.2 m in the north-west to 1.7 m in the east (Landy et al., 2017). Around Churchill, the ice usually starts forming in October-November and breaks up in May-June (Gagnon and Gough, 2005, 2006). Since 1996 the open water season has, on average, increased by 3.1 (±0.6) weeks in Hudson Bay, with mean shifts in dates for freeze-up and break-up of 1.6 (±0.3) and 1.5 (±0.4) weeks accordingly (Hochheim and Barber, 2014).

There have been few studies of zooplankton community composition in Hudson Bay. Among the macrozooplankton species found in Hudson Bay, *Parsagitta elegans* is the most abundant species, followed by *Aglantha digitale* as the second most abundant (Estrada et al., 2012). The mesoplankton community in Hudson Bay is dominated by small copepods*: Oithona similis, Oncaea borealis*, and *Microcalanus* (Estrada et al., 2012). Zooplankton diversity is generally low at high latitudes (Conover and Huntley, 1991). Typically, salinity gradients and freshwater discharge play an important role in determining species diversity (Witman et al., 2008). Seasonality in food availability is another significant challenging factor for zooplankton in high latitudes (Bandara et al., 2016; Carmack and Wassmann, 2006; Varpe, 2012).

### 3. Data Collection and Methods

#### 3.1 Mooring Configuration and Set up

A bottom-anchored oceanographic mooring (Fig. 1b) was deployed at 109 m depth ~190 km north-east from the port of Churchill (59° 58.156' N 91° 57.144' W) 26 September 2016 and recovered on 30 October 2017. The mooring setup consisted of (i) one upward-looking 5-beam Signature 500 ADCP by Nortek placed at 38 m depth, (ii) upward-looking 4-beam 300 kHz Workhorse Sentinel ADCP by RD Instruments placed at 106 m depth and (iii) one Gurney Instrument "Baker Type" sequential sediment trap (Baker and Milburn, 1983) at 85 m with collection area of 0.032 m$^2$. Several conductivity-temperature, conductivity-temperature-turbidity and temperature-turbidity sensors were also deployed at various depths on the mooring, but the data obtained by these sensors were not analyzed in this study.

The velocity and acoustic backscatter (ABS) intensity were measured by RDI ADCP between 8 and 100 m at 2 m depth intervals, with a 15-min ensemble time interval and 15 pings per ensemble. The ADCP velocity measurement precision and resolution were ± 0.5% and ± 0.1 cm s$^{-1}$, respectively. The accuracy of the ADCP vertical velocity measurements are not validated, however, the RDI reports that the vertical velocity is more accurate, by at least a factor of two than the horizontal velocity (Wood and Gartner, 2010). The compass accuracy was ± 2° and compass readings were corrected by adding magnetic declination.

The sediment trap was programmed to start a collection at 4 October 2016 0:00 CST with intervals of 35 days for each vial collected. Prior to boarding the vessel, sediment trap preservative density solution was prepared at the Churchill Northern Studies Centre (CNSC). To prepare the solution, 10 L of seawater was collected from the Churchill port wharf and filtered through 0.7 μm Whatman GF/F filters. The salinity of the filtered seawater was adjusted from 26.7 to 37 psu with 88.065 g of ultra-clean sea salt. Borax (44.4 g) was slowly added to 0.45 L of 37% formaldehyde, placed on a magnetic stir plate overnight to dissolve, and decanted into 8.55 L of filtered seawater. Approximately 1 hour before deployment of the sediment traps, pre-acid cleaned vials were placed inside the pre-programmed sampling carrousel and filled to the surface with the preservative solution. The trap was assembled and kept upright prior to and during deployment. During deployment, the different species of zooplankton were captured by the sediment trap (Fig. 6).

#### 3.2 Data Collection and Post Processing

ADCPs, unlike echo-sounders (Lemon et al., 2012, 2001), are limited in deriving accurate quantitative estimates of biomass due to calibration difficulties because their acoustic beams are narrow and inclined from the vertical (Brierley et al., 1998; Lemon et al., 2008; Sato et al., 2013; Vestheim et al., 2014). But with the application of beam geometry correction, ADCPs are commonly used for qualitative studies, as they can provide information on zooplankton presence and behaviour (Hobbs et al., 2014; Last et al., 2016; Petrusevich et al., 2016). To correct for the ADCP beam geometry, we derived the volume backscatter strength (VBS) $S_v$ in dB from echo intensity following the procedure described by *Deines*, (1999). The issue of

acoustic signal scattering by bubbles, waves and sea ice was addressed by removing the top 8 m readings from all backscatter and velocity data.

The total sky illumination for day and night was modelled using *skylight.m* function from the astronomy package for Matlab (Ofek, 2014) and a simple exponential decay radiative transfer model for estimating under ice illumination (Grenfell and Maykut, 1977; Perovich, 1996). Transmittance through the sea ice was calculated following Eq.(1):

$$T(z) = (1 - \alpha)e^{-k_t z}, \tag{1}$$

where $\alpha$ is the surface albedo, $\kappa_t$ is the bulk extinction coefficient of the sea ice cover, and $z$ is the ice thickness. The values of the coefficients used in the exponential decay model were adjusted for the first-year sea ice: $\alpha = 0.8$ and $\kappa_t = 1.2$. We did not have any data for snow cover available, so a presence of the snow cover was omitted in the transmittance model. However, an albedo of 0.8 was used to simulate the high albedo at visible wavelengths for snow-covered or white ice surfaces.

The thickness of (Figure 2) ice at the mooring location was estimated from the ice draft evaluated from the distance to the ice-ocean interface measured by the Nortek ADCP(Banks et al., 2006; Björk et al., 2008; Shcherbina et al., 2005; Visbeck and Fischer, 1995). The draft was further transformed to the ice thickness by multiplying with a factor of 1.115 for density difference between seawater and sea ice (Bourke and Paquette, 1989). The acoustic-derived thicknesses were corrected for ADCP tilt, sea surface height and atmospheric pressure (Krishfield et al., 2014) and for the speed of sound. The extreme outliers were excluded, and the mean daily ice thicknesses were calculated for further analysis (Figure 2).

The Environment and Climate Change Canada weather station at Churchill Airport (YYQ) located ~190 km south-west from the mooring location provided wind data for most of the time of mooring deployment, except for the period of March 27 – April 7, 2017. The daily mean wind speed magnitude was used to compile the wind speed time series (Figure 3c).

On recovery of the mooring, sediment trap samples were photographed poured into acid cleaned 250 mL amber glass bottles and stored in the dark at approximately 4 °C during transport to the Centre for Earth Observation Science, University of Manitoba. Samples were poured through 500 µm NITEX mesh sieve to separate the larger zooplankton fraction. 500 µm mesh was selected to maintain consistency and allow for comparison with previous studies (see Forbes et al., 1992; Pospelova et al., 2010). Because of this, smaller species, nauplii, eggs and fecal pellets were largely missed from the >500 µm fraction. However, the >500 µm organisms represent the group of zooplankton primarily detected as ADCP backscatter (Cisewski and Strass, 2016; Pinot and Jansá, 2001). Zooplankton taxonomy identification was conducted at the Freshwater Institute (DFO) to the lowest taxonomic level possible, enumerated and measured. The entire sample was scanned for large and rare organisms and then the sample was split, with a Motoda box splitter, and a minimum of 300 organisms was counted for each sample.

## 4. Results

### 4.1. Ice Thickness and Under-ice Illumination.

At the mooring location, the ice started rapidly forming in the second week of December. By mid-December thickness reached 0.4 m and was gradually growing till the middle of March up to 1 m (Figure 2). Afterwards, the ice thickness at the mooring location varied due to seasonal factors, e.g. polynyas, sea ice melting, etc.

Modelled under-ice illumination time series, as well as the volume backscatter strength and vertical velocity time series, were presented in the form of actograms (Figures 3d-g and 4). An actogram, being a common method of data display in chronobiological research, has recently been used for displaying zooplankton DVM (Hobbs et al., 2018; Last et al., 2016; Petrusevich et al., 2016; Tran et al., 2016).

The actogram of the modelled under-ice illumination (Figure 3i and 4e) shows continuous daily maximums at noon with minimum values of 2000 lux around the winter solstice and reaching maximum values of 10000 lux in the middle of summer. Maximum under ice lunar illumination was around 0.1 lux during full moon under sea-ice about 0.5 m thick.

### 4.2 Volume Backscatter Strength (VBS)

For analyzing the depth-dependent behaviour of scatterers involved in diurnal vertical migration, we computed the volume backscatter strength (VBS) time series at noon (Figure 3a) and at midnight (Figure 3b). The mean difference between noon-time and midnight VBS was ~9±1dB at 96-100m depth layer and -3dB±1 at 10-28 m layer. Running F-statistic test returned statistical significance with 95% confidence for VBS difference below 58 m and above 48 m. Noon-time series show persistent maximum backscatter strength near the bottom below 92 m depth, which is consistent with DVM. Some scatter stayed at noon at 60-80 m layer during October-January and at 70-80 m in June-July.

The near-bottom maximum for the midnight time series of VBS is significantly less compared to that for noon. Midnight time series during October-February and May-July showed a wider spread of scatterers over the depth. During winter months (December - February), the thickness of this layer of midnight bottom scatterers gradually decreased with the growth of sea ice. There are periods of higher VBS at the bottom layer with the same periodicity of 14 days as $M_2$ and $S_2$ tidal components superposition maxima (spring tide) throughout the whole time series. There was a seasonal variation of these periodic VBS maxima: they were increasing during summer-fall and decreasing in winter. It should be noted that during November-January there were higher values of backscatter below 80 m depth.

VBS was calculated for depths of 8, 20, 60, 80 and 92 m and is shown as actograms in Figure 3d-h. Overall, VBS actograms show a similar shape to that of the under-ice solar illumination actogram (Figure 3i). This resemblance in shape is outlined by reduced VBS at 8 and 20 m actograms (Figure 3d-e) and enhanced at 60, 80 and 92 m actograms (Figure 3f-h) during dawn and dusk. Reduced under ice illumination from December to March corresponded with reduced VBS through the whole water column, followed by increased illumination during ice breakup and open water periods (April to October) and an increase in

VBS within all five depth bins. Like the noon and midnight VBS time series, there is a relatively higher signal at 60, 80 m and 92m depth in November-January during the night.

The VBS actograms (Figure 3d-h) show the presence of vertical bands of higher VBS with 14 days periodicity at multiple depths. In the upper 8 and 20 m (Figure 3d-e), these bands are spreading through the night period, while at 80 and 92 m actograms the bands spread throughout the whole day with different values of VBS during the day and night. In the 8 m actogram (Figure 3d) there are also non-periodic bands of high backscatter that span from 1 to 5 days in duration. These bands spread throughout the whole day and correspond with the periods of wind speed increasing to strong wind, gale and storm values (30 km/h and up) during the ice-free season (Figure 3c).

Figure 3c shows daily mean wind speed measured at Churchill airport (YYQ). There were observed several periods of mean wind speed higher than 30 km/h, which corresponds with strong wind (37-61 km/h) and gale (62-87 km/h) wind speed values, with maximum wind gusting up to 77 km/h. Normally these storm events lasted from 1 to 6 days.

## 4.3 Vertical velocity actograms

The vertical velocity actograms were calculated for the same depths as VBS actograms (Figure 4a-d). Positive velocities are associated with the upward movement of particles. The seasonal shape of vertical velocity actograms is similar to the shape of under-ice illumination (Figure 3c and 4e) and VBS actograms (Figure 3d-g). The change in vertical speed associated with spring tide is present on the vertical velocity actograms in a form of slanted strips of 14-day periodicity, with amplitude increasing with depth and reaching maximum values in the range of 10-15 mm/s.

The vertical velocity actograms were post-processed (Figure 4f-i) to remove the semi-diurnal tidal components ($M_2$ and $S_2$) from the vertical velocity data which otherwise would create a tidal background signal in a form of slanted strips of 14-day periodicity on the actograms (Figure 4a-d). A tidal harmonic analysis was performed for the vertical velocity time series using T_Tide toolbox for Matlab (Pawlowicz et al., 2002). There was a small distinguishable diurnal variation of vertical velocity in 20 and 60 m actograms (Figure 4f and g) during the period of the full moon in October, November and December resembling the slanted shape of lunar illumination at the under-ice illumination actogram (Figure 4e).

## 4.4 Wavelet analysis

Time series of the wavelet power spectrum for the semidiurnal tidal currents were computed for accounting their spring-neap and seasonal variability. Wavelet for horizontal and vertical velocities (Figure 5b and c) show absolute maximum values during spring tides, which is consistent with the full moon and new moon phases (Figure 5a). The power spectrum range for horizontal velocities was in general, over one order higher than for vertical velocity, which is consistent with the fact that horizontal tidal currents tend to be at least an order of magnitude larger than vertical ones. There is a spacial difference between horizontal and vertical velocities power spectrum. The horizontal velocities wavelet has maximums that spread through the whole water column during the ice-free season, and below 30 m depth in the presence of ice cover (December-April). The vertical velocities

spectrum during October-April has maximums mostly concentrated below 70 m depth. There is a seasonal variation for the vertical velocity wavelet with May-June wavelet maximums started spreading through the whole water column.

For the analysis of ADCP measured current velocities, we used wavelet transformation to derive the time-dependent behaviour of horizontal and vertical current velocities at the semi-diurnal tidal frequency band that dominates the backscatter spectrum. In this study, we used the generalized Morse wavelet (with parameters $\beta=100$ and $\gamma=3$) and jWavelet toolbox (part of jLab toolbox) for signal processing (Lilly, 2017, 2019; Lilly and Gascard, 2006; Lilly and Olhede, 2009).

### 4.5 Sediment trap zooplankton

Zooplankton >500 μm captured in the sediment trap samples (Figure 6) were dominated (>98%) by five taxa including two calanoid copepods (*Calanus glacialis* and *Pseudocalanus* spp.), a pelagic sea snail (*Limacina helicina*), a gelatinous arrow worm (*Parasagitta elegans*) and an amphipod (*Themisto libellula*) (Table 1, Figure 7). The abundance of organisms in the trap was generally lowest from March to July with the exception of juvenile (2 mm length) *T. libellula* in bottle 6.

## 5. Discussion

### 5.1 Zooplankton Species Associated with DVM in Hudson Bay

The presence of seasonal ice cover acts as a barrier to using traditional zooplankton sampling techniques. But using both moored or ice-tethered ADCPs in high latitudes had been successful for studying zooplankton presence, behaviour and particularly DVM patterns (Darnis et al., 2017; Hobbs et al., 2018; Petrusevich et al., 2016; Wallace et al., 2010). Even though acoustic backscatter from the single-frequency ADCP does not provide any information on the identity of zooplankton species involved in DVM but signal strength can provide an indication of zooplankton presence provided there is information on the zooplankton species. Sound is effectively scattered by objects of the size of the wavelength. For 300 kHz ADCP, it is about 5 mm. It is known that zooplankton species with body size less than the wavelength by an order of magnitude (in our case 0.5-5mm) are capable of creating strong backscatter when there is a sufficient abundance of them in the water column (Cisewski and Strass, 2016; Pinot and Jansá, 2001). The backscatter strength of zooplankton species also depends on their acoustic properties, such as shape, internal structure, orientation in the water column and body composition, that causes a difference between the speed of sound in their bodies and surrounding seawater (Stanton et al., 1994, 1998a, 1998b). For example, the species with hard shells (like *Limacina helicina*) and gaseous enclosures scatter sound stronger than gelatinous ones (Lavery et al., 2007; Warren and Wiebe, 2008). It should be mentioned that 300kHz ADCP can be effectively used for suspended sediment transport monitoring (Venditti et al., 2016), but here are some general considerations that need to be taken into account. 300 kHz ADCPs were used for suspended sediment monitoring mostly in the rivers with high sediment loads (hundreds of mg L$^{-1}$). Our mooring was located ~190 km north-east from the Churchill river which does not create a significant plume of sediments into the system. The mooring turbidity sensor located at 41m depth did not record values higher than 34 FTU which corresponds with TSS of ~30 mg/L, with average turbidity of 7 FTU which corresponds with TSS ~5 mg/L. At

100 m depth, we do not expect high levels of sediments from resuspension. Also taking into consideration the fact that that sound is effectively scattered by objects of the size of the wavelength and that the mean particle size detected by 300 kHz ADCP is in the range of 0.5 to 5 mm (Jourdin et al., 2014), sporadic smaller scatterers, like sediments, phytoplankton, etc. can be effectively eliminated as potential scatterers. This allows us to consider zooplankton as the main scatterers in our case.

Fish also can be detected with the ADCP used. It should be noted though that large mesopelagic fishes are rare in the Canadian Arctic (Berge et al., 2015a). Arctic cod (*Boreogadus saida*) is the dominant pelagic fish in the Canadian Arctic (e.g. Benoit et al., 2008; LeBlanc et al., 2019) therefore the acoustic signals related to fish are generally assumed to be only Arctic cod. The distribution of Arctic cod is known for regions such as the Beaufort Sea (Geoffroy et al., 2016) and Baffin Bay (LeBlanc et al., 2019). However, there is little known for Hudson Bay. It is expected that Hudson Bay Arctic cod behave similarly, with adult aggregations near the bottom in deep waters and young (year 1/2) and larval stages in surface aggregations. The young cod are ice-associated during the winter period, i.e., no migration to depth. As such, any backscatter associated with near-surface young cod would have been removed as part of the removal of the top 8 m of backscatter during post-processing. Arctic Cod do not school. So, its presence in the proximity of the mooring will be more sporadic and acoustic backscatter will be significantly less than the backscatter from much more abundant zooplankton.

The trap samples reflect the presence of >500 μm zooplankton in the water column during the annual cycle. However, species absent from the trap samples (e.g., *L. helicina* in January-March) does not validate absence from the water column. The most abundant species from the zooplankton trap catch (*Parasagita elegans*, *Pseudocalanus* and *L. helicina*) had lengths of 20-30 mm, 0.6-1.4 mm and 0.4-2 mm respectively. Less abundant species from the trap (*Calanus glacialis* and *Themisto libellula*) had lengths of 2.8-4.2 mm and 7.2-31.8 mm, respectively. *P. elegans* and *T. libellula* lengths are in the range of ADCP wavelength and thus should effectively act as scatterers. Lengths of *C. glacialis, Pseudocalanus* and *L. helicina* are less than the wavelength by an order of magnitude. However, their abundance in the water column during open water season (Estrada et al., 2012) is high enough (>1000 ind m$^3$) to expect a backscatter signal. *L. helicina's* hard shell should be another contributing factor to backscatter strength. Therefore, we assume that all the species identified in the sediment trap could act as acoustic scatterers contributing to the VBS signal analyzed in this study

The zooplankton caught in our sediment trap provide general information on the zooplankton community composition and its change over the course of the year near the mooring location. Sediment trap samples may not quantitatively reflect zooplankton composition in the water column due to species-specific collection efficiencies. Comparisons between net and trap samples from Franklin Bay indicate that the abundance of *L. helicina* and some species of copepods could be estimated from sediment traps whereas the abundance of other key species, such as *C. hyperboreus*, could not be accurately estimated from sediment trap samples (Makabe et al., 2016).

The ADCP analyses indicate that zooplankton in Hudson Bay undergo both seasonal and diel migration. This is similar to measured seasonal migration by copepod species in the southern Arctic Ocean and in Rijpfjorden in Svalbard (Falk-Petersen et al., 2008). Seasonal migration is occurring in Hudson Bay despite shallower overwintering waters than in Svalbard and the Beaufort Sea. The observed diel migration in Hudson Bay is similar to other Arctic locations (Berge et al., 2014, 2015a; Hobbs

et al., 2018; Last et al., 2016; Petrusevich et al., 2016) suggesting that DVM is an important consideration for carbon/nitrogen transfer within the relatively shallow Hudson Bay system.

Zooplankton species identified from the sediment trap suggest that multiple species could be involved in the DVM. The identification of individual species involved in DVM is not currently possible and is challenged by issues such as the overlapping of signals. Comparison between acoustic and net data in Kongsfjorden, Svalbard led to the conclusion that the acoustic backscatter signal from numerically dominant *Calanus* copepods is typically overwhelmed by the signal from larger and less abundant zooplankton species, such as *Themisto* (Berge et al., 2014). Large copepods (like *Calanus* spp.) and chaetognaths (*P. elegans*) were observed performing diel migrations in Kongsfjorden (Darnis et al., 2017). While our sediment trap showed the prevalence of gelatinous zooplankton species (Fig. 7 – *P. elegans*), but the detection of their migration by ADCP backscattering could be underestimated because gelatinous species are weak scatterers.

Regardless, there is a pump of carbon/nitrogen occurring within Hudson Bay based on zooplankton DVM, and seasonal differences (discussed in the next section) could impact this vertical transport of elements. The collected acoustic data at hand are not valid to quantify zooplankton biomass involved in DVM. However, we can use them to document and understand better important aspects of DVM, such as links between its seasonal cycle and dynamics of sea-ice cover and under-ice illuminance, and the effects of wind storms and tides on DVM patterns.

**5.2 DVM seasonal cycle, sea-ice cover and under-ice illuminance**

The mooring site is located 6° south of the Arctic circle and polar twilight zone. Hudson Bay located more south than other seasonally sea-ice covered Arctic and sub-Arctic regions where DVM was observed. In those locations, DVM during the winter was primarily controlled by twilight and the lunar light (Last et al., 2016; Petrusevich et al., 2016). In this study, DVM was generally controlled by solar illumination throughout the whole year, which is evident from the shape of VBS (Figure 3d-h) and vertical velocity actograms (Figure 4). The actograms are nearly symmetric around astronomic midnight (dashed horizontal line, Figures 3 and 4), and winter and summer solstice. During dawn and dusk, there was reduced VBS at 8 and 20 m actograms (Figure 3d-e) and enhanced at 60, 80 and 92 m actograms (Figure 3f-h). These dawn and dusk absences and enhancements can be interpreted as an indication of zooplankton swimming behaviour during these periods, following nocturnal DVM pattern. The increased backscatter at dawn and dusk at 60 and 80 m actograms was observed regardless of the presence of ice cover.

The noon-time VBS time series showed consistent maximum backscatter strength below 92 m depth (Figure 3a). Compared to the midnight time series (Figure 3b), it is clear that the backscatter was associated with DVM rather than sediment resuspension caused by the lunar semi-diurnal $M_2$ tide with a period of 12 hours 25 minutes. The midnight VBS time series (Figure 3b) and VBS actograms (Figures 3d-h) confirm that the zooplankton were aggregated in the upper water column at midnight, likely feeding.

Seasonal variations in zooplankton migration and distribution in the water column were observed throughout the entire time series. The sediment trap at 85 m depth may have captured zooplankton species migrating vertically and possibly also

individuals sinking to the bottom (Figure 6). The strong VBS of –70dB during noon at the 90-100 m depth layer (Figure 3a), compared with –80dB at midnight (Figure 3b), suggests that noon-time DVM-associated zooplankton biomass was primarily located at the bottom layer through the annual cycle. From October to the middle of January, however, there was a layer of VBS in the range of –80 to –75dB at 60-80m depth, which can be interpreted that some of the zooplankton were staying at that depth instead of migrating all the way down to the bottom for daytime or to the surface at night. The 60-80 m aggregation of zooplankton, from October to January, corresponds with the first three sampling bottles of the sediment trap when there was the highest abundance of zooplankton observed with the abundance of dominant species per 35-day sampling period, decreasing from 720 down to 250 ind m$^{-3}$ (Figure 7). From the middle of January to early May, most of the zooplankton biomass at midnight did not migrate above 60 m depth. From May to July zooplankton returned to the vertical migration pattern observed from date to date when zooplankton remained near the bottom at noon and migrates to the surface at night. In July, some zooplankton stayed in the surface layer at noon. This corresponds to the beginning of the ice-free season (Figure 2) when long daylight and abundance of phytoplankton disrupts DVM. Once the sea ice was completely gone in early August, there was a change in zooplankton distribution in the water column. During midnight, some zooplankton remained at the bottom and while others migrated to the surface layer likely feeding during the short night and moving back down to the bottom for the light time. This suggests that different zooplankton scatter species and/or size classes are responding differently to both solar cues and ice cover.

In certain cases vertical velocity actograms can be used for estimating swimming direction and velocity (Petrusevich et al., 2016) when for estimation of swimming direction actograms are averaged for layers of several meters depth and for velocity estimation individual profiles were averaged over a period of few days. This method works well when there is no tidal signal to be subtracted from the vertical velocity data, otherwise, it makes computation rather complicated.

**5.3 Masking of DVM signal in the upper layer by storms**

The 8 m depth actogram (Figure 3d) shows several bands of higher VBS of different durations, that are not observed at the deeper layers. These bands spread throughout the entire 24-hour day for a duration of one to several days. These bands (Figure 3d) nicely correspond with daily mean wind speed exceeding 25 km/h (Figure 3c) during most of the ice-free season (October-mid December 2016 and September-October 2017). Irregular spots of higher VBS can be related to the bubbling generated by the wind forcing. In contrast, during the ice-covered season, periods of high winds did not associate with higher VBS. For example, on 7-10 March 2017, the daily mean wind was to 66 km/h, but there were no bands of higher VBS at the 8 m actogram (Figure 3d), indicating that ice cover partly protected the water column from wind stress. Irregular spots of higher VBS (Figure 3d) during the ice-covered period (February-March) could be attributed to the frazil ice formation. With the onset of spring melt (May-July), there is also more noise-type VBS that could be attributed to the release of the ice-rafted sediments during the melting of the sea ice. The large amount of sediment present in the May-July sediment trap bottles (Figure 6) provides proof for the presence of sinking sediment during this period.

An alternative explanation of higher VBS at 8 m depth is a different feeding pattern for non-visual predators like chaetognaths (including *P. elegans*). While mature species are known to perform DVM, in some cases juvenile individuals were found near the surface during the daytime (Brodeur and Terazaki, 1999).

## 5.4 Disruption of DVM by the spring tide

Time series of the wavelet power spectrum for horizontal and vertical velocities (Figures 5b, c) show absolute maximum values during spring tides, which correspond to full moon and new moon phases (Figure 5a). For 92 m depth, the 14-day running correlation (Figure 5d, green line) between midnight VBS (blue line) and vertical velocity wavelet (red line) was calculated. Correlations exceeding ±0.53 are statistically significant at the 95% confidence level (Figure 5d, yellow shading). Pink shading identifies the events when this statistically significant positive correlation was observed. Negative correlations are artificial and have no physical meaning. The periods of low correlation were from the end of November to mid-January, mid-February to mid-March, April to mid-June and the first half of September. A statistically significant positive correlation suggests the relationship between VBS and tidal forcing.

In the presence of background stratification, the barotropic tide interacts with sloping bottom topography in the proximity of the mooring location (Figure 1), which is typical for Hudson Bay (Petrusevich et al., 2018). This interaction generates vertical divergence and convergence of tidal flow, resulting in the depth-dependent behaviour of the vertical velocity at a tidal frequency here defined as the baroclinic tide. The seasonal character of the baroclinic tide can also be affected by density stratification. During May-October 2017 the vertical velocity wavelet maximums were amplified (Figure 5c). During this period there were DVM disruptions throughout the water column that are clearly evident on VBS actograms (Figures 3d-g) and at noon VBS time series (Figure 3a).

Zooplankton normally avoid spending additional energy to cross such an interface a horizontal interface with a strong velocity gradient, thereby resulting in a weakened or absence of a DVM signal (Petrusevich et al., 2016). Similar observations of disrupted zooplankton vertical migration had been linked to upwelling and downwelling events (Dmitrenko et al., 2019). The same considerations can be applied to this study when water dynamics are impacted by vertical currents generated by the baroclinic tides and enhanced during spring tide. During spring tide, zooplankton showed a weakened DVM to avoid moving against the vertically diverging and converging tidal flow, as follows from the VBS actograms and correlation between time series of VBS and vertical velocity wavelet. This disruption can be moon controlled as those reported by Hobbs et al. (2014); Last et al. (2016), and Petrusevich et al. (2016). However, in this study, the lunar origin of this disruption is attributed to the tidal dynamics rather than the moonlight, because disruptions occurred during the full moon and new moon phases.

## 6. Conclusion

A one-year-long acoustic backscatter and vertical velocity time series, obtained using a 300 kHz ADCP on a mooring deployed from September 2016 to October 2017 in south-east Hudson Bay (~190 km north-east from the port of Churchill), revealed a distinct diurnal pattern consistent with zooplankton diel vertical migration (DVM).

In this study, we were able to identify that the presence of multiple zooplankton species that could have been involved in DVM from samples collected by the sediment trap. The sediment trap was programmed to collect settling material over a complete annual cycle (35-day interval and averaging period) and consequently, the collection was not timed to shorter tidal cycles. This limited the identification of the specific species whose DVM was detected by the 300 kHz ADCP and altered by $M_2$ tidal water dynamics. Using shorter sediment trap time intervals or in situ sampling required for the identification of the zooplankton species involved in DVM in future mooring deployments.

The major factors determining the observed DVM pattern were as follows:

*Illuminance.* Unlike other ice-covered and ice-free Arctic and sub-Arctic locations such as Svalbard and north-east Greenland (Last et al., 2016; Petrusevich et al., 2016), DVM in Hudson Bay is controlled by solar illumination throughout the whole year, not by moonlight.

*Tidal dynamics.* The tide in Hudson Bay is mostly lunar semidiurnal ($M_2$) with an amplitude of about a few meters. The area in the proximity of the mooring has variable bottom topography (Figure 1). The barotropic tide interacts with bottom topography generating tidal flow diverging and converging vertically. It seems that zooplankton tends to avoid spending additional energy swimming against the vertical flow. This response of zooplankton is consistent with the zooplankton tendency to stay away from the layers with enhanced water dynamics and to adjust its DVM accordingly.

*Storm induced disruptions.* When daily mean wind speed exceeded 25 km/h during most of the ice-free season in the surface layer there were observed irregular spots of higher VBS related to the bubbling generated by the wind forcing.

**Data and code availability**

The backscatter and velocity data are archived in the Centre for Earth Observation Science (University of Manitoba) and are restricted for open access in accordance with University of Manitoba policy during two years after the observations completed. The Matlab code used for data processing is available from VP upon request.

**Authors contribution**

VP prepared the manuscript with contributions from all co-authors (ID, AN, SK, MK, ZK, DB and JE). VP, SK and MK deployed and retrieved the mooring in Hudson Bay. AN processed and presented zooplankton data from the sediment trap. VP processed and presented acoustic data from the ADCP. SK processed and provided data for the ice draft.

## Competing interests

The authors declare that they have no conflict of interest.

## Acknowledgements

Funding support from BaySys (Hudson Bay System Study), the Natural Sciences and Engineering Research Council (NSERC), Discovery Grant program, and the ArcticNet Network of Centres of Excellence made field data collection and analysis possible. We also acknowledge contributions from the Canada Excellence Research Chair (CERC) and Canada Research Chair

(CRC) programs to support the University of Manitoba team. We would like to give special thanks to Alexis Burt of Fisheries and Oceans Canada for processing zooplankton taxa. We would like to thank the Captain Neil J. MacDonald, Chief Officer Kevin Jones and crew of the CCGS Henry Larsen, the Canadian Coast Guard, technician Sylvan Blondeau of Laval University and Christopher Peck of the University of Manitoba for their assistance with successful mooring retrieval, as well Nathalie Thériault for coordinating BaySys field logistics.

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

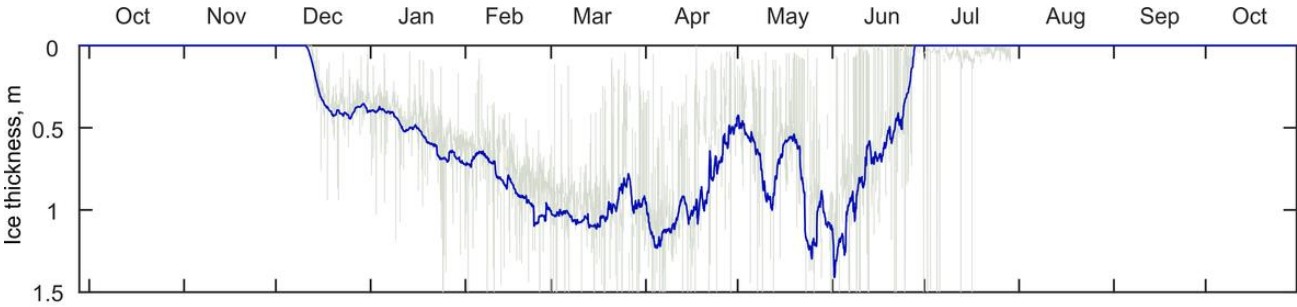

**Figure 1. a) The bathymetric map of the Hudson Bay region and location of the mooring (AN01). The inset map shows Hudson Bay on the map of Canada. b) Schematic illustration of the mooring AN01 setup.**


**Figure 2. ADCP-measured ice thickness at the mooring location (AN01) during winter 2016-2017. Gray and blue lines represent the filtered and daily averaged ice thicknesses, respectively.**

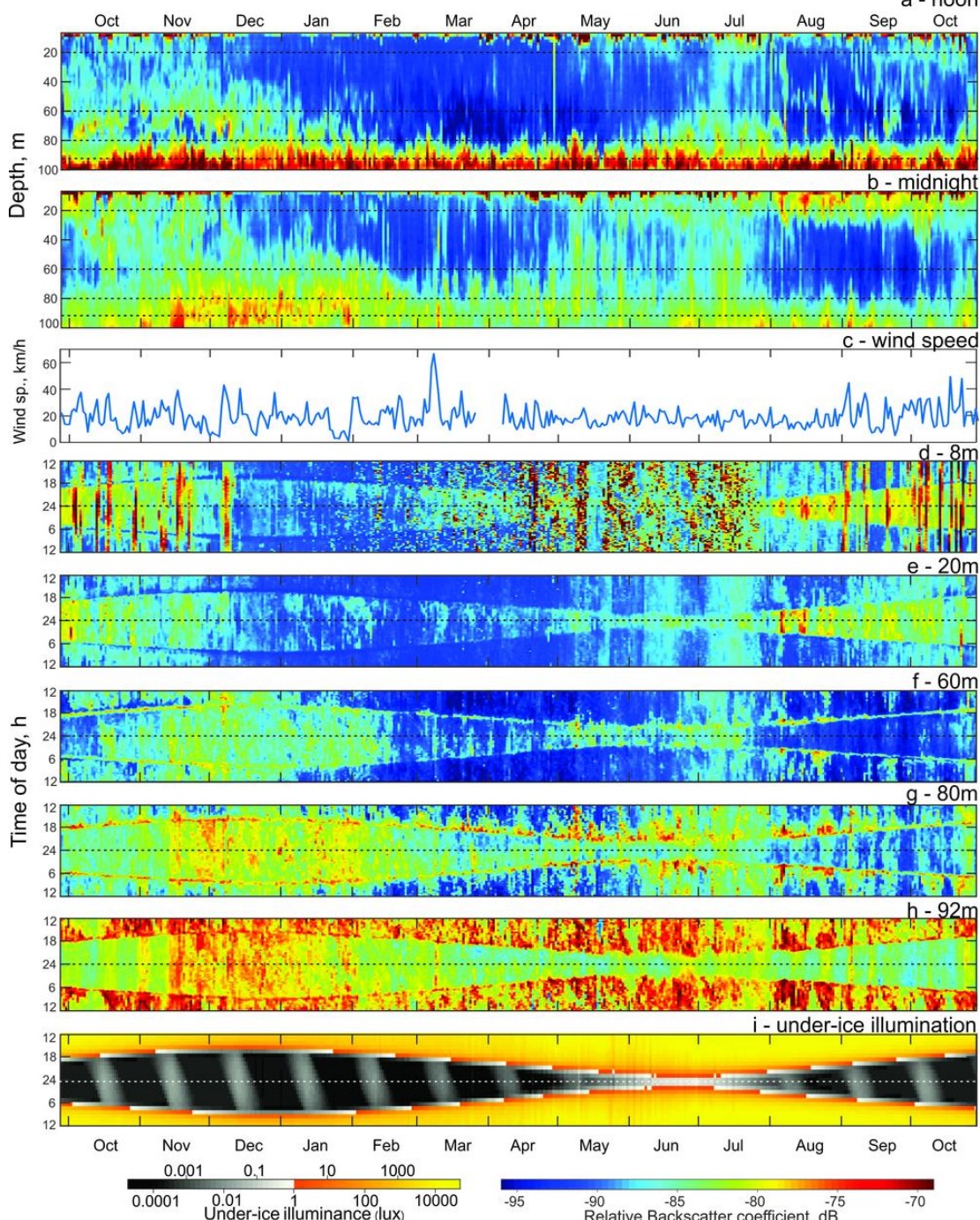

**Figure 3.** Time series (October 2016 to October 2017) of the (a) ADCP acoustic volume backscatter coefficient at noon and (b) at midnight, (c) daily mean wind speed measured at Churchill airport (YYQ), and (d-i) actograms of ADCP acoustic backscatter at five depth levels: (d) 8m, (e) 20 m, (f) 60 m, (g) 80 m and (h) 92 m and (i) modeled under-ice illuminance. Dashed horizontal lines represent the astronomical midnight. The diurnal signal is presented at the vertical axis, while the long-term changes in diurnal behaviour are presented along the horizontal axis.

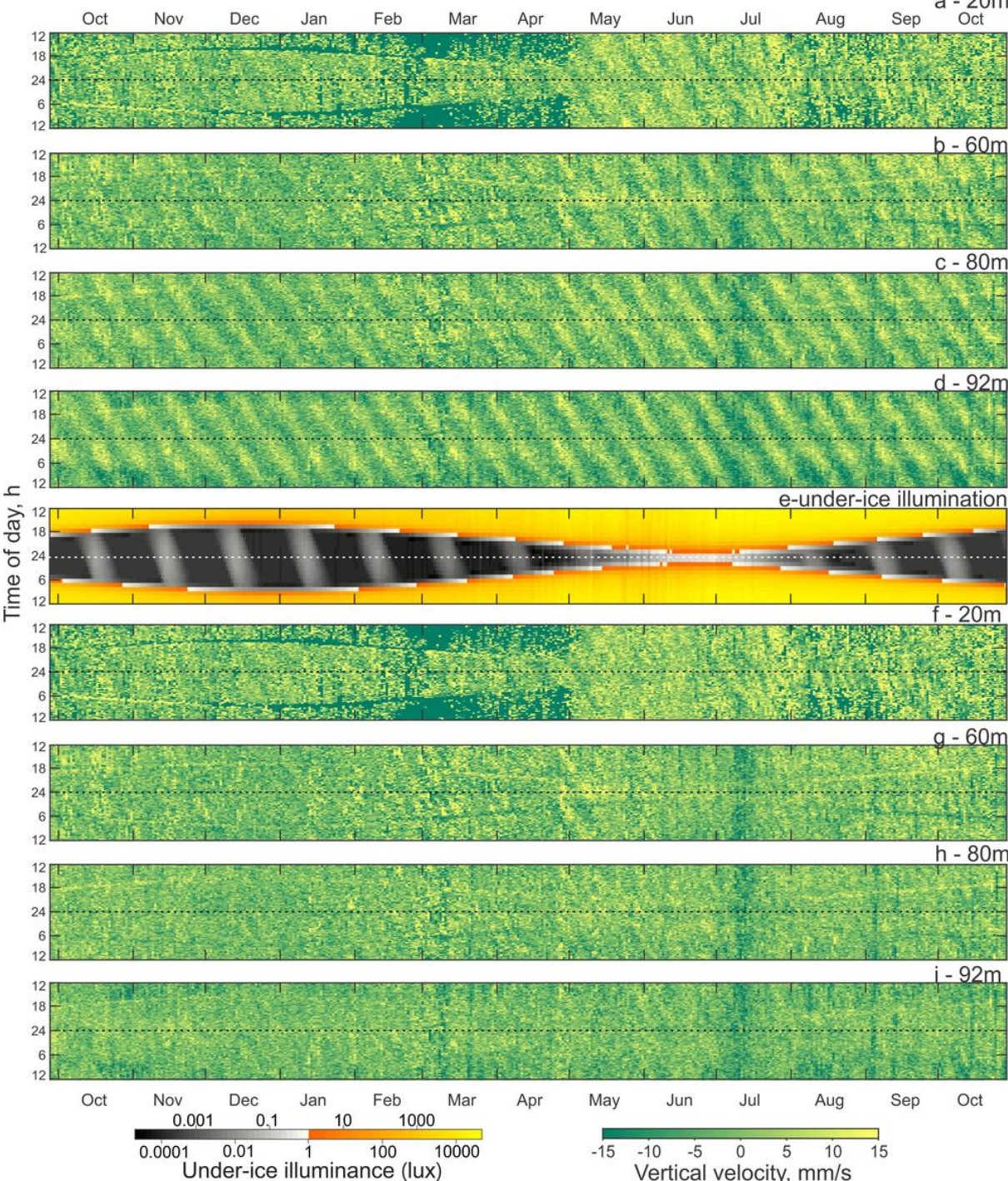

Figure 4. Actograms of (a-d) ADCP-measured vertical velocity (mm/s) at four depth levels: (a) 20 m, (b) 60 m, (c) 80 m and (d) 92 m, (e) modelled under-ice illuminance and (f-i) residual vertical velocity (mm/s, tidal signal subtracted) at four depth levels: (f) 20 m, (g) 60 m, (h) 80 m and (i) 92 m. Positive/negative values correspond to the upward/downward net flux. Dashed horizontal lines represent the astronomical midnight.

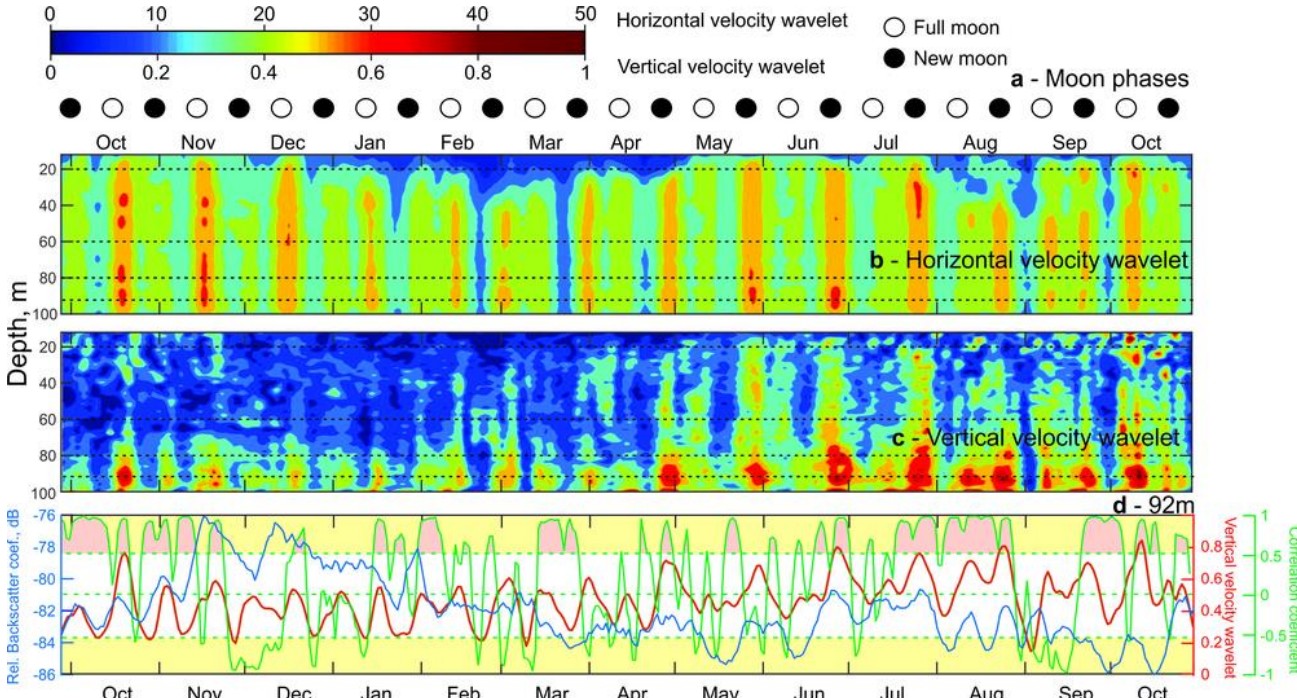

Figure 5. Time series of (a) lunar phases for 2016 – 2017. ○ - Full moon. ● – New moon. (b) and (c) - the absolute value of wavelet power spectrum for the time series of horizontal velocity (b) and vertical velocity (c) computed for semi-diurnal frequency band (12 h) as a function of depth. (d) – the correlation coefficient (green line) between time series of VBS (blue line) and vertical velocity wavelet (red line) at 92 m depth. Yellow shading identifies the correlation coefficient levels exceeding ±0.53, which are statistically significant for the 95% confidence. Pink shading identifies the events when this statistically significant correlation was observed.

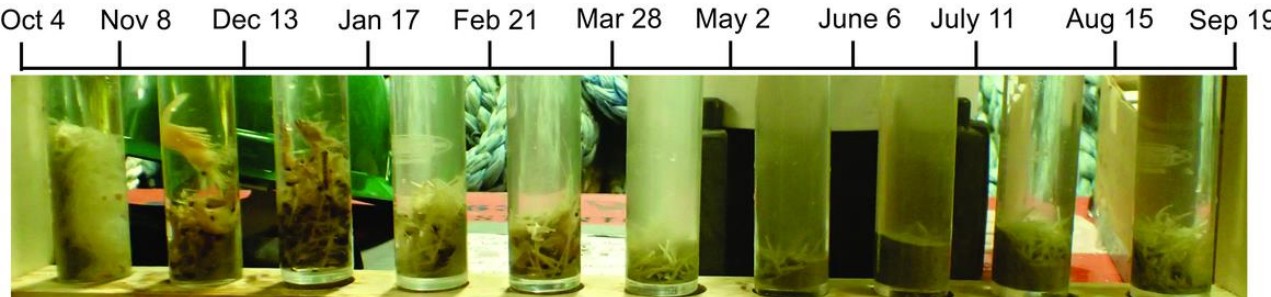

Figure 6. Contents of the sediment trap for ten 35-day intervals.

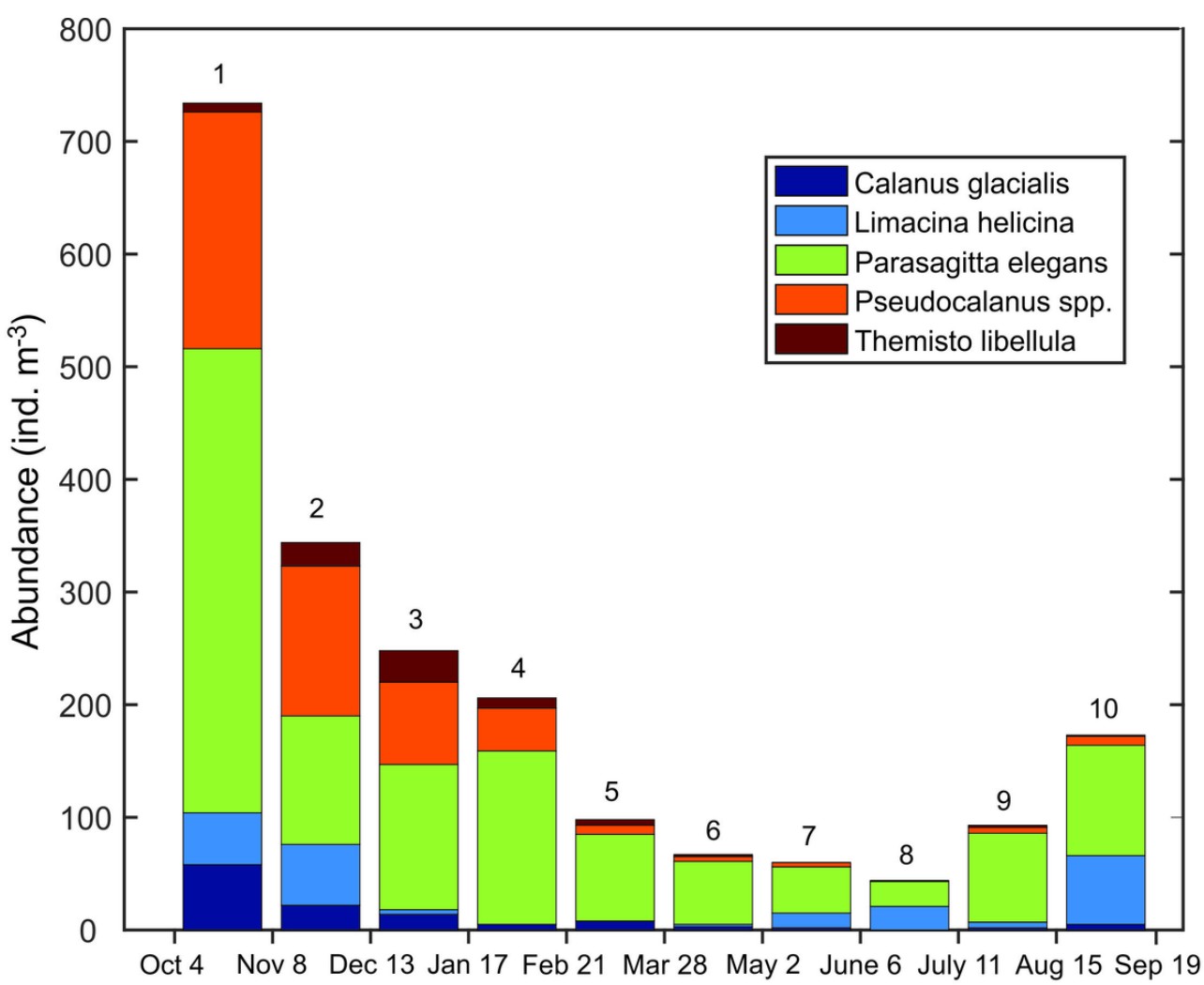

**Figure 7. Abundance (ind. M⁻³) of dominant zooplankton (>500 µm) in each bottle of the sediment trap at AN01, October 2016 to August 2017.**