# Peer review of "Impact of tidal dynamics on diel vertical migration of zooplankton in Hudson Bay"

_Ocean Science, 2019_

## Referee Comment (RC1) · Anonymous Referee #1 · 7 Nov 2019

**Review comments to 'Impact of tidal dynamics on diel vertical migration of zooplankton in Hudson Bay' by Petrusevich et al.**

In this work the authors use ADCP backscatter and velocity data from a mooring in western Hudson Bay to infer diel vertical migration patterns. They present some interesting data, however, the results presentation and interpretation require more in depth work to make this work publishable.

General comments:

1. How do you make sure that the backscatter that the ADCP sees is actually zooplankton? The ADCP will 'see' anything from suspended sediment, bubbles, krill, fish, plankton etc. The presence of increased amounts of backscatter within the water column together with location in the water column and timing can give suggestions as to what is causing the backscatter: e.g. storminess leads to an increase in bubbles, or the day vs night timing gives an indication that you are seeing vertical movement of biological matter. BUT, how can you infer that you are seeing zooplankton moving rather than fish that migrate vertically from your data? Your sediment traps are fairly small and could easily miss swimming species.

2. What particle size classes does your ADCP 'see'? How does this tie in with your definition of zooplankton?

3. The analysis and description of the results needs to go into more depth. At the moment it is mostly descriptive and does not go into enough detail describing the very interesting dataset. Additionally, part the results are presented in the discussion. This needs to be tidied up.

4. This is a paper supposedly on DVM, however, in the results DVM patterns aren't really described at all, nor is the tidal modulation.

5. The authors also need to be careful to separate out the supposed swimming behaviour of the zooplankton, i.e. DVM, from the tidally induced movement of particles (also backscattering) in the water column.

6. The authors need to take more care in backing up their claims with either features that can be seen in the their data or else from relevant pieces of work in the literature.

Specific comments:

Introduction: How do you define zooplankton here? What size classes or types of plankton are included or not included?

Line 20-22: Better off before the question at the end of the paragraph

Lines 24-26: There are other diel migration patterns as well. See e.g. introduction to Cisewski et al, 2010

Line 30: Would it worth looking at Antarctic literature as well?

Line 60: There are multiple sets of references and an extra 1.

Line 94: Given that you talk about stratification in the introduction and there are some interesting vertical signals in your plankton distributions analysing stratification could lead to some interesting additional conclusions. In plots 3 a and b there is some indication that plankton distribution could be linked to MLD.

Line 101 -109: How representative are the samples from your sediment trap at capturing actual distributions of zooplankton in the water column? I guess you are assuming that organisms 'fall' into the trap and cannot avoid it by swimming back out of the funnel?

Lines 110 to 116: They are also limited because you cannot tell what is causing the backscatter! How do you determine that your backscatter is zooplankton and not other biological matter or sediment…?

Line 116: Full stop missing and 'by multiplying 1.115' → by multiplying with a factor of 1.115. I assume this is to correct for density differences between water and ice? If yes, say so.

Line 135-140: How do you know that the >500um makes up the greatest amount of backscatter? Why do you choose a >500um mesh? Mesoplankton is normally defined as 200um to 2mm in size…. What is the ratio between smaller species and larger species?

Line 145: Do you have evidence for the reasons the ice varied? If yes, detail it, if not, I would speculate here.

Figure 2: I find the y-axis very unintuitive – I feel it would be best to flip it upside down with 0 m at the bottom and 1.5 m at the top

Section 4.1: Section lists 'wind data' however it is not mentioned in this section.

Figure 4. The green-yellow colorscale makes it really hard to see anything. I suggest changing it a blue white red colorscale.

Lines 156: Maximum backscatter is consistent with many things. It only becomes consistent with DVM once you compare it to the midnight timeseries.

Section 4.2: Could you back your descriptive results up with some numbers? Calculate the difference in backscatter between day and night in the different layers – is there a statistical difference?

Lines 156-157: Evidence of MLD?

Line 165: are → is, actograms → actogram

Line 165-166: Describe the resemblance in shape – when in the day-night cycle do you see increased backscatter? Mention dawn and dusk enhancements/absences which could be indicative of swimming behaviour

Figure 3. I would add a time series of sea ice cover here for ease of comparison.

Also Figure 3. In the 80 m and 60m band there is increased backscatter at dawn and dusk regardless of ice cover.

Also Figure 3. Do positive vertical velocities resemble an upward movement of particles?

Lines 180 and after: I would make it clear somewhere here that your vertical velocities encompass both moving of water particles but also other particles in the water column and are thus a mixture of

both. Really importantly, the velocities are a mixture of passive (tidally-driven) and actively moving particles (e.g. zooplankton or possibly fish).

Lines 181-183:

> You say they have the same shape, but what does that actually mean? What signals do you see in the velocities? E.g. in the top layers you see negative vertical velocities in the 20 m layer at dusk and positive ones at dawn. However, assuming that positive vertical velocities resemble an upward movement of particles, this means that there is a net downward migration at dusk – this would be counterintuitive to the DVM you are describing, at least for this layer.

> Can you back any of your claims with numerics? Yes, you can see a pattern but is it statistically significant? E.g. calculate mean backscatter in daylight hours vs night hours

Lines 185-187: This description belongs in the methodology.

Lines 187-190: So what is the diurnal variation? What is the pattern you see? How does this match up with your results in Figure 2? There you seem to find the strongest signals in the deeper layers?

Lines 191 to 192: It is not clear to me what you have done here. Are you looking at semi-diurnal horizontal tidal currents? Or are you looking at horizontal tidal currents from the ADCP which are semi-diurnally dominated? How have you post-processed your current data? Have you removed any tidal signals? What bandwidth are you using? It looks like it is greater than the semi-diurnal frequency?

Line 200-204: This belongs in the methodology. What do these parameters mean for your data?

Line 200-201: I don't see a semi-diurnal signal in the currents – this is a spring-neap signal

Lines 194-195: Yes, horizontal tidal currents tend to be at least an order of magnitude larger than vertical ones.

Section 4: It is not quite clear to me what the point of this exercise is? You show that you have stronger horizontal and vertical currents during spring tides and the opposite for neap tides. This is commonly known. So where does the link to DVM come in? Where do you show the tidal modulation your title promises? And you have removed the 12 hr signal that would give you the DVM? In Hudson Bay, the tides vary throughout the year due to changes in ice cover and stratification – how do you separate these effects from DVM?

Figure 5. Describe the results from panel 5d in the text. How do you calculate your correlation coefficient? What time window is used? How did you obtain your backscatter time series? You obtain both significant positive and negative correlations, why are the negative ones not shaded pink?

Line 214: Provided you know you are looking at zooplankton – see general point 1

Line 217: How do you know it's 5 mm – what is the size range on either size? What uncertainty exists here? What about objects larger than 5 mm? Or sediment?

Line 224: You are assuming that your trap is reflective of what's in the water column. Did you check this or can you prove this is correct any other way?

Line 262-237: This needs to be backed up with an appropriate description of your results

Lines 258 – 280: This is not discussion but description of results – move to results section

Lines 294-300: This belongs in the results

Line 301: Where do you show the water column is stratified?

Line 301-304: You get changes in vertical velocity without interactions with topography? You don't show there are interactions with topography here. Also, the paper you reference here is for South Hudson Bay…

Line 305-307: This can also be caused by the presence or lack of sea ice (see. e.g. Kleptsova and Pietrzak, 2018, Ocean Modelling); also reference needed

Line 312-313: You don't show that there is reduced DVM during spring tide

Line 332-333: You do not show this anywhere.

Line 336-337: Again, you speculate this, there is no evidence for this presented.

---

## Referee Comment (RC2) · Anonymous Referee #2 · 16 Nov 2019

The paper is very well written and with a clear and concise message. I have a few comments / questions:

1) line 55, objective 3: Why is not solar light mentioned here?

2) How do you separate actively migrating from passively sinking (dead) organisms?

3) Line 125. Ice thickness measured by ADCP - does there exist any groundtruthing data for this method? There are no references provided, except one that does not seem to be relevant? This needs to be updated / clarified

4) There is a basic understanding or basis for a DVM pattern regulated by light that is not really presented, but which is essential to the entire manuscript. I would strongly suggest that the authors first describe this general and consistent DVM, and then focus

on how this is disrupted. One way of doing this would be to compared noon with midnight mean position in the water column throughout the entire data series.

When these issues are sorted, I recommend that the manuscript is accepted for publication

---

## Referee Comment (RC3) · Anonymous Referee #3 · 26 Nov 2019

General

This work exploits acoustic data from an ADCP moored over an annual cycle, backed with zooplankton identification in sediment trap samples, to document the seasonal dynamics of zooplankton DVM at a seasonally ice-covered site in Hudson Bay. The data analyses sound complete but the highlighted results do not seem particularly novel in the way they are presented. Maybe one approach to deal with this perception is to work on a more thorough comparison of the patterns observed in Hudson Bay with other regions. More in-depth interpretation of the linkages between the acoustic observations and zooplankton biology would also help this work. In particular, tidal effects on DVM seem to be emphasized by the title but this does not appear that well in the Discussion.

[Figure]

The structure of the manuscript needs to be better strengthened as there are pieces of different sections that should belong to other ones, as detailed in the specific comments. The title takes into account only one aspect addressed by this work.

There is also an important issue that should be addressed either in the Introduction of the Discussion: is the trap a valid way to identify the scatterers?

Specific comments

Title

The title does not reflect the scope of this work properly since the tidal effects was only one part of the Discussion

Abstract

Line 13-14: Give the information on potential migrators instead of telling that they could be identified.

Line 14: "migrating scatters"? what does that mean? How can a scatter migrate?

Introduction

Line 20: I would remove "synchronized" from the sentence, as DVM doesn't have to be synchronized to transport C and N to depth. Furthermore, "synchronized" is used in the following sentence that explains DVM.

Line 28: Explain better why this question needs to be addressed

Line 39: remove "to" after "help"

M&M

Line 79: It is "Macrozooplankton" we are talking about here and not "Microzooplankton"

Line 80: "Parasagitta" instead of "Sagitta"

Lines 83-85: This information does not fit in here in the description of the study area.

The authors should find a more proper place to use it if needed. The whole paragraph on zooplankton should be moved somewhere else.

Line 92: the sampling area of this trap is very small and may cause a bias in zooplankton catching toward the smaller individuals that need to be addressed.

Line 137: a citation is needed to back the information on the size fraction effectively sampled by the ADCP

- "Motoda" instead of "Motodo"

Results Line 143: Does that mean that in a matter of a few days, the ice thickness reached 0.4 m?

Line 145: remove "the"

Line 146: replace "were" by "are"

Line 154: replace "scatters" by "scatterers", here and elsewhere. This sentence is a piece of the Methods and would fit better in the previous section

Line 156: the part on DVM in this sentence is interpretation of Results and would fit better into the Discussion.

Line 158: statistics?

Line 160: "midnight bottom scatters layer" by "layer of midnight bottom scatterers"

Line 162: "maxima" instead of "maximums"

Line 164: remove "observed"

Line 166: "shape shows a similar overall shape..." too many "shape" and "overall" here

Line 205: remove brackets

Line 208: It is "libellula", not "libellua"

Line 208-209: This sentence does not provide results information.

Discussion

The first paragraph of a discussion should give justice to the Results and novel knowledge provided by the work and entice the reader to learn more about the issue. I would turn the first sentence differently so that it would not look so much like it emphasizes the weakness of the ADCP-based method to study zooplankton patterns.

Line 215: Studies like the one by Makabe et al (2016) address the issue of the usefulness of sediment trap samples for the description of zooplankton community composition and seasonal change by comparing zooplankton caught in sediment traps with ones sampled by plankton nets. What is found in the trap samples does not necessarily give a good picture of the zooplankton composition in the water column. The trap might miss the importance of scatterers that are not well sampled by the small-aperture trap. Themisto might be quite under sampled by the small trap. Furthermore, traps are known to oversample pteropods that stop swimming and sink when they touch the mooring line. Some change in behavior influencing the depth range of zooplankton will also have an impact on trap catching efficiency. This has to be kept in mind and mentioned.

Makabe, R., Hattori, H., Sampei, M., Darnis, G., Fortier, L., Sasaki, H., 2016. Can sediment trap-collected zooplankton be used for ecological studies? Polar Biol., doi:10.1007/s00300-00016-01900-00307.

Line 235: DVM patterns have already been documented in another part of Hudson Bay (Runge and Ingram 1991). The authors should give credit to the pioneer study in this paragraph.

Runge, J.A., Ingram, R.G., 1991. Under-ice feeding and diel migration by the planktonic copepods Calanus glacialis and Pseudocalanus minutus in relation to the ice algal production cycle in southeastern Hudson Bay, Canada. Mar. Biol. 108, 217-225.

Line 245: Well, we do not have the elements of information yet to tell if this pump is important or not. Importance would depend on the real scatterers, and depth and stratification state of the water column.

Line 246: "DVM" and not "DMV"

Line 247: "vertical transport of elements" instead of "vertical energy transfer"

Line 248: I don't think that it is worth introducing the next sections in that way. Normally a logical suite of sub-sections should be enough.

Or replace by something like : "the acoustic data at hand are not valid to quantify zooplankton biomass involved in DVM. However, we can use them to document and understand better important aspects of DVM, such as: links between its seasonal cycle and dynamics of sea-ice cover and under-ice illuminance, and the effects of wind storms and tides on DVM patterns".

Line 252: "south" instead of "southern location". In any case, this sentence should be rewritten to improve its clarity. Make the message straighter.

In general, there are too many figure citations in this section. If the Results section is clearly written, there is no need to cite those figures again. The Discussion should take on from the Results described in the previous section.

Line 271: by definition, the trap does not measure abundance but a rate of capture or sinking in the case of inert particles. Thus, I fear that it can be too misleading to use the term "abundance" in that case even though it is mentioned that it is the abundance in the trap sample after 35 days of opening. This is because the rate will not necessarily be related to the abundance of organisms in the water column. This is a tricky issue that should be addressed carefully.

Line 280: one alternate explanation that should be discussed is that of different feeding patterns. Some non-visual predators like chaetognaths might not need to move that much if their zooplankton prey change their migration patterns as well etc..

Line 281: Is it disruption of masking of the DVM signal? From the interpretation, it is not possible to understand if the storms act on the zooplankton responsible for the DVM patterns, or if other physical action produce backscatter that prevent the visualization of DVM. The paper should relate storms to zooplankton behavior or change the title of this sub-section, which then would much less relevant.

Line 288: remove "present"

Line 291: "amount of" and not "amount in"; "provides" and not "provide"

---

## Editor Comment (EC1) · Mattias Green (Editor) · 8 Jan 2020

The three reviews, provided by experts across the disciplines covered in the paper, raises a series of relevant points that needs to be addressed in a revised version. I don't have anything extra to add myself, but I particularly want to see the stronger link to the tide addressed (it is in a special issue for tides after all), and a further motivation to ensure that what you see really are zoo plankton.

---

## Author Comment (AC1) · 30 Jan 2020

**Response to review comments to 'Impact of tidal dynamics on diel vertical migration of zooplankton in Hudson Bay' by Anonymous Referee # 1**

We highly appreciate helpful comments and suggestions from Anonymous Referee #1. In the following, *the comments by the reviewer are in italics* and our responses to the comments are in normal characters. The revised manuscript text is underlined. **The line numbering (in bold)** is referenced to the marked-up manuscript version.

*In this work the authors use ADCP backscatter and velocity data from a mooring in western Hudson Bay to infer diel vertical migration patterns. They present some interesting data, however, the results presentation and interpretation require more in depth work to make this work publishable. General comments:*

*1. How do you make sure that the backscatter that the ADCP sees is actually zooplankton? The ADCP will 'see' anything from suspended sediment, bubbles, krill, fish, plankton etc. The presence of increased amounts of backscatter within the water column together with location in the water column and timing can give suggestions as to what is causing the backscatter: e.g. storminess leads to an increase in bubbles, or the day vs night timing gives an indication that you are seeing vertical movement of biological matter. BUT, how can you infer that you are seeing zooplankton moving rather than fish that migrate vertically from your data? Your sediment traps are fairly small and could easily miss swimming species.*

Thank you for pointing out the limitations of using ADCP and sediment trap combination for studies of zooplankton, but in presence of seasonal ice cover and logistic difficulties associated with using traditional zooplankton sampling techniques using both moored or ice-tethered ADCPs in high latitudes had been successful for studying zooplankton presence, behaviour and particularly DVM patterns (Darnis et al., 2017; Hobbs et al., 2018; Petrusevich et al., 2016; Wallace et al., 2010).
For better zooplankton detection, it is recommended to use dual-frequency ADCP. However, in our study, we were able to overcome issues of backscatter from fish by applying post-processing techniques of the backscatter and presenting it in the form of actograms. This approach allowed us to present the portion of backscatter that is associated with DVM related to zooplankton.

Below we would like to clarify the possible scatterers suggested by the reviewer.

1) **Suspended sediments.** While it is true that 300kHz ADCP can be used for suspended sediment transport monitoring (Venditti et al., 2016), there are some general considerations that need to be taken into account. 300 kHz ADCPs were used mostly in the rivers with high sediment loads (hundreds of mg/L). The mooring was located ~190 km north-east from the Churchill River, which does not create a significant plume of sediments in the system. Our mooring turbidity sensor, located at 41m depth, did not record values higher than 34 FTU, which corresponds with TSS of ~30 mg/L, with an average 7 FTU or TSS about ~5 mg/L. At 100 m depth, we do not expect a high level of sediments from resuspension. Another consideration is the fact that sound is effectively scattered by objects of the size of the wavelength. For 300 kHz ADCP, it is about 5 mm (lines 235-241). Thus, smaller scatterers, like sediments, phytoplankton, etc. can be effectively eliminated as potential scatterers. This allows us to consider zooplankton as the main scatterers in our case.

2) **Bubbles**: interference of bubbles was addressed by removing the top 8 m of surface readings. This removes the interference of waves and sea ice. The surface 8 m was removed from all data. This has been clarified in the methods:

**Lines 132-133:** The issue of acoustic signal scattering by bubbles, waves and sea ice was addressed by removing the top 8 m readings from all backscatter and velocity data.

3) **Fish.** This is a valid concern regarding the fact that fish may be detected by ADCP. It should be noted though that large mesopelagic fishes are rare in the Canadian Arctic (Berge et al. 2015). Arctic cod (*Boreogadus saida*) is the dominant pelagic fish in the Canadian Arctic (e.g. Benoit et al. 2008, LeBlanc et al. 2019) such that acoustic signals related to fish are generally assumed to be only Arctic Cod. The distribution of Arctic Cod is known for regions such as the Beaufort Sea (Geoffroy et al. 2016) and Baffin Bay (LeBlanc et al. 2019). However, little is known for Hudson Bay. It is expected that Hudson Bay Arctic Cod behave similarly, with adult aggregations near the bottom in deep waters and young (year 1/2) and larval stages in surface aggregations. The young cod are ice-associated during the winter period, i.e., no migration to depth. As such, any backscatter associated with near-surface young cod would have been removed as part of the removal of the top 8 m of backscatter during post-processing. Arctic Cod do not school. So, its presence in the proximity of the mooring will be more sporadic and acoustic backscatter will be significantly less than the backscatter from more abundant zooplankton.

4) **Sediment trap.** Another point of concern was about the use of sediment trap in this study. It is true that the sediment traps do not tell us anything about the upward or horizontal movement of zooplankton. The trap captures only what is falling/swimming to the bottom. The trap samples generally indicate what zooplankton species are present in the study area and how the relative proportion of captured zooplankton changed over the duration of the year. Recently an issue of the usefulness of sediment trap samples for description of zooplankton composition and seasonal change was addressed by Makabe et al., 2016. They compared zooplankton caught in sediment traps with the ones sampled by plankton nets. Pteropods (*L. helicina*) showed a good correlation between sediment trap and plankton nets sampling. *C. glacialis* showed good correlation with environmental variables and not with net-collected abundance.

We revised our manuscript in this matter accordingly:

**Lines 282-287:** The zooplankton caught in our sediment trap provide general information on the zooplankton community composition and its change over the course of the year near the mooring location. Sediment trap samples may not quantitatively reflect zooplankton composition in the water column due to species-specific collection efficiencies. Comparisons between net and trap samples from Franklin Bay indicate that the abundance of *L. helicina* and some species of copepods could be estimated from sediment traps whereas the abundance of other key species, such as *C. hyperboreus*, could not be accurately estimated from sediment trap samples (Makabe et al., 2016).

Benoit D, Simard Y, Fortier L. 2008. Hydroacoustic detection of large winter aggregations of Arctic cod (Boreogadus saida) at depth in ice-covered Franklin Bay (Beaufort Sea). J Geophys Res 113(C6): 9 p. doi:10.1029/2007jc004276.
Berge, J., Renaud, P. E., Darnis, G., Cottier, F., Last, K., Gabrielsen, T. M., Johnsen, G., Seuthe, L., Weslawski, J. M., Leu, E., Moline, M., Nahrgang, J., Søreide, J. E., Varpe, Ø., Lønne, O. J., Daase, M. and Falk-Petersen, S.: In the dark: A review of ecosystem processes during the Arctic polar night, Prog. Oceanogr., 139, 258–271, doi:10.1016/j.pocean.2015.08.005, 2015.

Geoffroy M, Majewski A, LeBlanc M, Gauthier S, Walkusz W, et al. 2016. Vertical segregation of age-0 and age-1+ polar cod (Boreogadus saida) over the annual cycle in the Canadian Beaufort Sea. Polar Biol 39(6): 1023-1037. doi:10.1007/s00300-015-1811-z.

Geoffroy, M., Robert, D., Darnis, G. and Fortier, L.: The aggregation of polar cod (Boreogadus saida) in the deep Atlantic layer of ice-covered Amundsen Gulf (Beaufort Sea) in winter, Polar Biol., 34(12), 1959–1971, doi:10.1007/s00300-011-1019-9, 2011.

LeBlanc, M., Gauthier, S., Garbus, S.E., Mosbech, A. and Fortier, L., 2019. The co-distribution of Arctic cod and its seabird predators across the marginal ice zone in Baffin Bay. Elem Sci Anth, 7(1), p.1. DOI: http://doi.org/10.1525/elementa.339

Makabe, R., Hattori, H., Sampei, M., Darnis, G., Fortier, L., Sasaki, H., 2016. Can sediment trap-collected zooplankton be used for ecological studies? Polar Biol., doi:10.1007/s00300-00016-01900-00307.

Venditti, J. G., Church, M., Attard, M. E. and Haught, D.: Use of ADCPs for suspended sediment transport monitoring: An empirical approach, Water Resour. Res., 52(4), 2715–2736, doi:10.1002/2015WR017348, 2016.

*2. What particle size classes does your ADCP 'see'? How does this tie in with your definition of zooplankton?*

At 300 kHz, mesozooplankton (0.2 – 20 mm) can be detected. Not all mesozooplankton have their peak detection at 300 kHz. For example, euphausiids peak detection may be closer to 200 kHz and pteropods closer to 400 kHz. However, 300 kHz is within the detection range for the key species observed in the sediment trap and known from previous zooplankton taxonomy assessments from the study area. This includes the dominant copepods (e.g., Cottier et al. 2006). We make no attempt to distinguish between groups of zooplankton or to estimate abundance/biomass from the ADCP data. The backscatter shows the general presence of zooplankton and associated movement/transfer. There is a number of papers by Stanton et al. (1994, 1998a, 1998b) addressing the target strength of different zooplankton species for different frequencies.

We revised the discussion section accordingly:

**Lines 240-250:** Even though acoustic backscatter from the single-frequency ADCP does not provide any information on the identity of zooplankton species involved in DVM but signal strength can provide an indication of zooplankton presence provided there is information on the zooplankton species. Sound is effectively scattered by objects of the size of the wavelength. For 300 kHz ADCP, it is about 5 mm. It is known that zooplankton species with body size less than the wavelength by an order of magnitude (in our case 0.5-5mm) are capable of creating strong backscatter when there is a sufficient abundance of them in the water column (Cisewski and Strass, 2016; Pinot and Jansá, 2001). The backscatter strength of zooplankton species also depends on their acoustic properties, such as shape, internal structure, orientation in the water column and body composition, that causes a difference between the speed of sound in their bodies and surrounding seawater (Stanton et al., 1994, 1998a, 1998b).

**Lines 252-271:** It should be mentioned that 300kHz ADCP can be effectively used for suspended sediment transport monitoring (Venditti et al., 2016), but here are some general considerations that need to be taken into account. 300 kHz ADCPs were used for suspended sediment monitoring mostly in the rivers with high sediment loads (hundreds of mg L$^{-1}$). Our mooring was located ~190 km north-east from the Churchill river which does not create a significant plume of sediments into the system. The mooring turbidity sensor located at 41m depth did not record values higher than 34 FTU which corresponds with TSS of ~30 mg/L, with average turbidity of 7 FTU which corresponds with TSS ~5 mg/L. At 100 m depth, we do not expect high levels of sediments from resuspension. Also taking into consideration the fact that that sound is effectively scattered by objects of the size of the wavelength and that the mean particle size detected by 300 kHz ADCP

is in the range of 0.5 to 5 mm (Jourdin et al., 2014), sporadic smaller scatterers, like sediments, phytoplankton, etc. can be effectively eliminated as potential scatterers. This allows us to consider zooplankton as the main scatterers in our case.

Fish also can be detected with the ADCP used. It should be noted though that large mesopelagic fishes are rare in the Canadian Arctic (Berge et al., 2015). Arctic cod (*Boreogadus saida*) is the dominant pelagic fish in the Canadian Arctic (e.g. Benoit et al., 2008; LeBlanc et al., 2019) therefore the acoustic signals related to fish are generally assumed to be only Arctic cod. The distribution of Arctic cod is known for regions such as the Beaufort Sea (Geoffroy et al., 2016) and Baffin Bay (LeBlanc et al., 2019). However, there is little known for Hudson Bay. It is expected that Hudson Bay Arctic cod behave similarly, with adult aggregations near the bottom in deep waters and young (year 1/2) and larval stages in surface aggregations. The young cod are ice-associated during the winter period, i.e., no migration to depth. As such, any backscatter associated with near-surface young cod would have been removed as part of the removal of the top 8 m of backscatter during post-processing.  Arctic Cod do not school. So, its presence in the proximity of the mooring will be more sporadic and acoustic backscatter will be significantly less than the backscatter from much more abundant zooplankton.

Cottier, Finlo R., Tarling, Geraint A., Wold, Anette, Falk-Petersen, Stig, ( 2006), Unsynchronised and synchronised vertical migration of zooplankton in a high Arctic fjord, Limnology and Oceanography, 51, doi: 10.4319/lo.2006.51.6.2586.Stanton, T. K., Wiebe, P. H., Chu, D., Benfield, M. C., Scanlon, L., Martin, L. and Eastwood, R. L.: On acoustic estimates of zooplankton biomass, ICES J. Mar. Sci. J. du Cons., 51(4), 505–512, doi:10.1006/jmsc.1994.1051, 1994.
Stanton, T. K., Chu, D. Z., Wiebe, P. H., Martin, L. V and Eastwood, R. L.: Sound scattering by several zooplankton groups. I. Experimental determination of dominant scattering mechanisms, J. Acoust. Soc. Am., 103(1), 225–235, doi:10.1121/1.421469, 1998a.
Stanton, T. K., Chu, D. Z. and Wiebe, P. H.: Sound scattering by several zooplankton groups. II. Scattering models, J. Acoust. Soc. Am., 103(1), 236–253, doi:10.1121/1.421110, 1998b.

*3. The analysis and description of the results needs to go into more depth. At the moment it is mostly descriptive and does not go into enough detail describing the very interesting dataset. Additionally, part the results are presented in the discussion. This needs to be tidied up.*
*4. This is a paper supposedly on DVM, however, in the results DVM patterns aren't really described at all, nor is the tidal modulation.*

We have revised our manuscript adding the general description of different DVM patterns in the intro and additional description of the observed DVM pattern in the results.

**Lines 30-35:** There are three general DVM patterns: (1) The most common one is *nocturnal* when zooplankton ascends around sunset and remains at upper depth during the night, around sunrise descending and remaining at depth during the day (Cisewski et al., 2010; Cohen and Forward, 2002). (2) Then there is a *reverse* pattern when zooplankton is ascending up at dawn and descending at dusk (Heywood, 1996; Pascual et al., 2017). And finally, (3) there is *twilight* DVM pattern when zooplankton is ascending at sunset, then descending around midnight, then again ascending and finally descending at sunset (Cohen and Forward, 2005; Valle-Levinson et al., 2014). This pattern sometimes is called *midnight sink*.

**Lines 315-320:** In this study, DVM was generally controlled by solar illumination throughout the whole year, which is evident from the shape of VBS (Figure 3d-h) and vertical velocity actograms (Figure 4). The actograms are nearly symmetric around astronomic midnight (dashed horizontal line, Figures 3 and 4), and winter and summer solstice. During dawn and dusk there was reduced VBS at 8 and 20 m actograms (Figure 3d-e) and enhanced at 60, 80 and 92 m actograms (Figure 3f-h). These dawn and dusk absences and enhancements can be interpreted as indication of zooplankton swimming behaviour during these periods, following nocturnal DVM pattern. The increased backscatter at dawn and dusk at 60 and 80 m actograms was observed regardless the presence of ice cover.

Tidal modulation is described in the section 5.4 Disruption of DVM by the spring tide with some minor revisions in the text.

*5. The authors also need to be careful to separate out the supposed swimming behaviour of the zooplankton, i.e. DVM, from the tidally induced movement of particles (also backscattering) in the water column.*

As we pointed there will be hardly any tidally driven sediment resuspension in the proximity of the mooring. The tides in Hudson Bay are semidiurnal, while DVM is diurnal signal with a "V-shaped" nocturnal pattern (Last et al., 2016) at Actograms. The superposition of two signals can be seen for example on vertical velocity actogram (Figure 4a-d), with tidal currents removed there is a remaining DVM signal with a "V-shaped" nocturnal pattern (Figure 4f-i), that matches V-shape of under-ice illumination actogram and VBS actograms (Figure 3d-h).

*6. The authors need to take more care in backing up their claims with either features that can be seen in the their data or else from relevant pieces of work in the literature.*

Thank you for this suggestion. We took this into consideration in the revised version and in responses to the specific comments below.

*Specific comments:*
*Introduction: How do you define zooplankton here? What size classes or types of plankton are included or not included?*:

Zooplankton are defined here as any individual greater than 500 µm. The zooplankton fraction was separated from the sediment trap sample by pouring the sample through 500 µm NITEX mesh sieve. Because of this, smaller species, nauplii, eggs and fecal pellets were largely missed from the >500 µm fraction. However, the >500 µm organisms represent the group of zoo-plankton primarily detected as ADCP backscatter (Cisewski and Strass, 2016; Pinot and Jansá, 2001). (**Lines 150-155**):

Zooplankton composition and diversity in the study region is addressed in Section 2: (lines 79-85 of the original submission. **Lines 93-100** in the revised version).

We added the following lines in the introduction:

**Lines 64-66:** In this study, we are focused on zooplankton organisms with sizes from 500 µm and up. This group of zooplankton primarily detected by ADCP backscatter (Cisewski and Strass, 2016; Pinot and Jansá,

2001) and allows comparison with previous studies on zooplankton caught by sediment traps (see Forbes et al., 1992; Pospelova et al., 2010).

*Line 20-22: Better off before the question at the end of the paragraph*

Thanks for the recommendation. We have revised and moved this sentence down.

**Lines 35-38:** DVM of zooplankton is an important process of the carbon and nitrogen cycle in marine systems, because it effectively acts as a biological pump, transporting carbon and nitrogen vertically below the mixed layer by respiration and excretion (Darnis et al., 2017; Doney and Steinberg, 2013; Falk-Petersen et al., 2008).

*Lines 24-26: There are other diel migration patterns as well. See e.g. introduction to Cisewski et al, 2010*

I have added a brief description with references for nocturnal, reverse and twilight migration patterns:

**Lines 30-35.** There are three general DVM patterns: (1) The most common one is *nocturnal* when zooplankton ascends around sunset and remains at upper depth during the night, around sunrise descending and remaining at depth during the day (Cisewski et al., 2010; Cohen and Forward, 2002). (2) Then there is a *reverse* pattern when zooplankton is ascending up at dawn and descending at dusk (Heywood, 1996; Pascual et al., 2017). And finally, (3) there is *twilight* DVM pattern when zooplankton is ascending at sunset, then descending around midnight, then again ascending and finally descending at sunset (Cohen and Forward, 2005; Valle-Levinson et al., 2014). This pattern sometimes is called *midnight sink*.

*Line 30: Would it worth looking at Antarctic literature as well?*

This is a good suggestion, but such a review of Antarctic literature would be more suitable for pan-arctic study, in this paper we are more looking into a regional sub-arctic region of Hudson Bay.

*Line 60: There are multiple sets of references and an extra 1.*

We have added the following references **(line 73-74)**: (Burt et al., 2016; Ingram and Prinseberg, 1998; Macdonald and Kuzyk, 2011; Petrusevich et al., 2018; St-Laurent et al., 2008; Straneo and Saucier, 2008).

*Line 94: Given that you talk about stratification in the introduction and there are some interesting vertical signals in your plankton distributions analysing stratification could lead to some interesting additional conclusions. In plots 3 a and b there is some indication that plankton distribution could be linked to MLD.*

That is a good point, but we have lack of actual multiple CTD sampling during that period, so we can only speculate about MLD without solid proof from oceanographic data from the mooring location.

*Line 101 -109: How representative are the samples from your sediment trap at capturing actual distributions of zooplankton in the water column? I guess you are assuming that organisms 'fall' into the trap and cannot avoid it by swimming back out of the funnel?*

Sediment traps were successfully used for qualitative assessment of the species responsible for acoustic backscatter (Berge et al., 2009; Cottier et al., 2006; Ota et al., 2008; Wallace et al., 2010; Willis et al., 2006, 2008). It is an accepted and recommended practice to simultaneously use samples collected by sediment traps to better under acoustic instruments backscatter signals (Makabe et al., 2016).

The sediment trap captures sinking zooplankton. There is a low probability that as the zooplankton sink they would reverse and search for a way out.  When compare to literature on zooplankton in Hudson Bay (see Estrada et al. 2012), the species collected within this sediment trap are representative of the zooplankton community we would expect to find in this system. Gelatinous zooplankton recorded by Estrada et al. 2012 also occurred in the zooplankton samples, however, at low numbers (data not shown). It is possible that gelatinous species are underestimated due to issues associated with preservation. This is noted at lines 221-226, 243-256 in the paper.

*Lines 110 to 116: They are also limited because you cannot tell what is causing the backscatter! How do you determine that your backscatter is zooplankton and not other biological matter or sediment…?*

For identification of biological swimmers, we are using special techniques like actograms, that allows us to distinguish diurnal and seasonal signal produced by biological swimmers from other possible passive scatterers.

Below we would like to clarify about the possible scatterers suggested by the reviewer.

1) **Suspended sediments.** While it is true that 300kHz ADCP can be used for suspended sediment transport monitoring (Venditti et al., 2016), there are some general considerations that need to be taken into account. 300 kHz ADCP were used mostly in the rivers with high sediment loads (hundreds of mg/L).  The mooring was located  ~190 km north-east from the the Churchill river which does not create a significant plume of sediments into the system. Our mooring turbidity sensor located at 41m depth did not recorded values higher than 34 FTU which corresponds with TSS of ~30 mg/L, with average 7 FTU or TSS about ~5 mg/L. At 100 m depth we do not expect high levels of sediments from resuspension. Another consideration is the fact that sound is effectively scattered by objects of the size of the wavelength. For 300 kHz ADCP, it is about 5 mm (lines 235-241). Thus, smaller scatterers, like sediments, phytoplankton, etc. can be effectively eliminated as potential scatterers. This allows us to consider zooplankton as main scatters in our case.

2) **Fish** may be detected with the ADCP used. It should be noted though that large mesopelagic fish are rare in the Canadian Arctic (Berge et al. 2015). Arctic cod (*Boreogadus saida*) is the dominant pelagic fish in the Canadian Arctic (e.g. Benoit et al. 2008, LeBlanc et al. 2019) such that acoustic signals related to fish are generally assumed to be only Arctic Cod. The distribution of Arctic Cod is known for regions such as the Beaufort Sea (Geoffroy  et al. 2016) and Baffin Bay (LeBlanc et al. 2019). However, little is known for Hudson Bay. It is expected that Hudson Bay Arctic Cod behave similarly, with adult aggregations near bottom in deep waters and young (year 1/2) and larval stages in surface aggregations. The young cod are ice associated during the winter period, i.e., no migration to depth. As such, any backscatter associated with near surface young cod would have been removed as part of the removal of the top 8 m of backscatter during post-processing.  Arctic Cod do not school. So, its presence in the proximity of the mooring will be more sporadic and acoustic backscatter will be significantly less than the backscatter from more abundant zooplankton.

*Line 116: Full stop missing and 'by multiplying 1.115' ▯ by multiplying with a factor of 1.115. I assume this is to correct for density differences between water and ice? If yes, say so.*

Corrected. The revised sentence:
**Lines 144-145:** The draft was further transformed to the ice thickness by multiplying with a factor of 1.115 for density difference between seawater and sea ice (Bourke and Paquette, 1989)

*Line 135-140: How do you know that the >500um makes up the greatest amount of backscatter? Why do you choose a >500um mesh? Mesoplankton is normally defined as 200um to 2mm in size…. What is the ratio between smaller species and larger species?*

This is valid point that meoszooplankton is normally defined as 200 um to 2 mm in size. All dominant taxa known from Hudson Bay are included in the >500 samples. Also, We chose >500 um mesh to maintain consistency and allow for comparison with previous studies (see Obrian, M.C. et al, 1991; Forbes et al; 1992; Posperlova et al 2010)

We have added this point to our revised manuscript:

**Lines 153-154:** 500 µm mesh was selected to maintain consistency and allow for comparison with previous studies (see Forbes et al., 1992; Pospelova et al., 2010).

Forbes, J. R., Macdonald, R. W., Carmack, E. C., Iseki, K., & O'Brien, M. C. (1992). Zooplankton retained in sequential sediment traps along the Beaufort Sea shelf break during winter. Canadian Journal of Fisheries and Aquatic Sciences, 49(4), 663-67
M.C. O'Brien, K. Iseki, R.W. Macdonald, J.R. Forbes, Yang Liangfeng, D. McCullough, 1991, Title
Can. Data Rep. Hydrogr. Ocean Sci.: 60, 238 pp
Pospelova, V., Esenkulova, S., Johannessen, S. C., O'Brien, M. C., & Macdonald, R. W. (2010). Organic-walled dinoflagellate cyst production, composition and flux from 1996 to 1998 in the central Strait of Georgia (BC, Canada): a sediment trap study. Marine Micropaleontology, 75(1-4), 17-37.

*Line 145: Do you have evidence for the reasons the ice varied? If yes, detail it, if not, I would speculate here.*

There is a paper titled "Atmospheric forcing drives the winter sea ice thickness asymmetry of Hudson Bay" prepared by our research group  and currently accepted for publication at JGR Oceans where the ice conditions and dynamics are addressed from ADCP measurements and remote sensing data. So, we preferred not to go deep into factors affected the ice variability in this particular paper dealing with DVM and tides.

Kirillov, S., Babb, D, Dmitrenko, I., Landy, D., Lukovich, J., Ehn, J., Sydor, K., Barber, D. and Stroeve, J. (2020) Atmospheric forcing drives the winter sea ice thickness asymmetry of Hudson Bay. Accepted @ J. Geophys. Res. Oceans

*Figure 2: I find the y-axis very unintuitive – I feel it would be best to flip it upside down with 0 m at the bottom and 1.5 m at the top*

This figure represents the ADCP-measured ice thickness calculated from the ice draft, so it sounds logical to represent it in this form.

*Section 4.1: Section lists 'wind data' however it is not mentioned in this section.*
Corrected
**Line 162:** 4.1. Ice Thickness and Under-ice Illumination

*Figure 4. The green-yellow colorscale makes it really hard to see anything. I suggest changing it a blue white red colorscale.*

We have tried different color schemes and this one (or just grayscale) seems to be the best for presenting actograms of ADCP-measured vertical velocity

*Lines 156: Maximum backscatter is consistent with many things. It only becomes consistent with DVM once you compare it to the midnight timeseries.*

We mentioned in the previous sentence, that in this section we are comparing noon time VBS with midnight (the following paragraph).

The revised paragraph:
**Lines 174-179:** For analyzing the depth-dependent behaviour of scatterers involved in diurnal vertical migration, we computed the volume backscatter strength (VBS) time series at noon (Figure 3a) and at midnight (Figure 3b). The mean difference between noon-time and midnight VBS was ~9±1dB at 96-100m depth layer and -3dB±1at 10-28 m layer. Running F-statistic test returned statistical significance with 95% confidence for VBS difference below 58 m and above 48 m. Noon-time series show persistent maximum backscatter strength near the bottom below 92 m depth, which is consistent with DVM. Some scatter stayed at noon at 60-80 m layer during October-January and at 70-80 m in June-July.

*Section 4.2: Could you back your descriptive results up with some numbers? Calculate the difference in backscatter between day and night in the different layers – is there a statistical difference?*

For reviewers' reference we generated a graph (below) of difference in VBS between noon and midnight (not included in the revised manuscript).

[Figure]

We have run F-statistics to find statistically significant difference and added to the revised manuscript:

**Lines 175-177:** The mean difference between noon-time and midnight VBS was ~9±1dB at 96-100m depth layer and -3dB±1 at 10-28 m layer. Running F-statistic test returned statistical significance with 95% confidence for VBS difference below 58 m and above 48 m.

*Lines 156-157: Evidence of MLD?*
Unfortunately, with our data (lack of CTD sampling at the mooring location) we can not positively confirm or deny the presence of MLD during that time.

*Line 165: are ⯈ is, actograms ⯈ actogram*
Thank you for pointing out this type. Got it corrected **(lines 187-188)**

*Line 165-166: Describe the resemblance in shape – when in the day-night cycle do you see increased backscatter?*

Added the following sentence:

**Lines 188-190:** This resemblance in shape is outlined by reduced VBS at 8 and 20 m actograms (Figure 3d-e) and enhanced at 60, 80 and 92 m actograms (Figure 3f-h) during dawn and dusk.

*Mention dawn and dusk enhancements/absences which could be indicative of swimming behaviour*

This is a valid observation. This increased backscatter can be explained by nocturnal migrating zooplankton passing through these layers at dawn and dusk.

We revised this section of manuscrtipt:

**Lines 317-320:** During dawn and dusk there was reduced VBS at 8 and 20 m actograms (Figure 3d-e) and enhanced at 60, 80 and 92 m actograms (Figure 3f-h). These dawn and dusk absences and enhancements can be interpreted as an indication of zooplankton swimming behaviour during these periods, following nocturnal DVM pattern. The increased backscatter at dawn and dusk at 60 and 80 m actograms was observed regardless the presence of ice cover.

*Figure 3. I would add a time series of sea ice cover here for ease of comparison.*
We were considering that option, but figure 3 was getting too big to fit the page, so we put figure 2 as a separate figure.

*Also Figure 3. In the 80 m and 60m band there is increased backscatter at dawn and dusk regardless of ice cover.*

This is a valid observation. This increased backscatter can be explained by nocturnal migrating zooplankton passing through these layers at dawn and dusk.

We revised this section of manuscrtipt:

**Lines 317-320:** During dawn and dusk there was reduced VBS at 8 and 20 m actograms (Figure 3d-e) and enhanced at 60, 80 and 92 m actograms (Figure 3f-h). These dawn and dusk absences and enhancements can be interpreted as an indication of zooplankton swimming behaviour during these periods, following nocturnal DVM pattern. The increased backscatter at dawn and dusk at 60 and 80 m actograms was observed regardless the presence of ice cover.

*Also Figure 3. Do positive vertical velocities resemble an upward movement of particles?*

Yes, positive vertical velocities are associated with the upward movement of particles.

We have added to the revised manuscript:

**Lines 204-205:** Positive velocities are associated with the upward movement of particles.

*Lines 180 and after: I would make it clear somewhere here that your vertical velocities encompass both moving of water particles but also other particles in the water column and are thus a mixture of both. Really importantly, the velocities are a mixture of passive (tidally-driven) and actively moving particles (e.g. zooplankton or possibly fish).*

This is valid comment and we are giving more explanation on usefulness of just vertical velocities for estimating biological swimmers below, particularly the challenge of the presence of a tidal signal.

We have added the following lines:

**Lines 344-347:** In certain cases vertical velocity actograms can be used for estimating swimming direction and velocity (Petrusevich et al., 2016) when for estimation of swimming direction actograms are averaged for layers of several meters depth and for velocity estimation individual profiles were averaged over a period of few days. This method works well when there is no tidal signal to be subtracted from the vertical velocity data, otherwise, it makes computation rather complicated

*Lines 181-183:*
*You say they have the same shape, but what does that actually mean? What signals do you see in the velocities? E.g. in the top layers you see negative vertical velocities in the 20 m layer at dusk and positive ones at dawn. However, assuming that positive vertical velocities resemble an upward movement of particles, this means that there is a net downward migration at dusk – this would be counterintuitive to the DVM you are describing, at least for this layer.*

In this section we presented vertical velocity actograms (showing presence of strong tidal signal) and vertical velocity actograms (cleared from the tidal signal) to show their general similarity of their seasonal shapes to VBS actograms, which attributes some of the signal to biological swimmers. There is a possibility that vertical velocity actograms could be used to estimate swimming direction and velocity, as we did in (Petrusevich et al., 2016), where we used vertical velocity actograms averaged for 14 m layers. This method works well in situation when there is no tidal signal to be subtracted, otherwise it makes the computation very complicated, especially in relation of propagation of errors.

Just for illustration purposes I attach some figures from that paper (from NE Greenland fjord, with low tidal dynamics) that deals with that method:

[Figure]

1 Nov (2013)   1 Dec   1 Jan   1 Feb   1 Mar   1 Apr   1 May (2014)

a

0   0.0001   0.01   1   100 lux

b

Cloud cover, %

Winter solstice

m04

c   16-30 m

d   40-54 m

Polar Night

m03

e   16-30 m

f   40-54 m

Cell Circulation

m02

g   16-30 m

h   40-54 m

Time of day, UTC h

1 Nov (2013)   1 Dec   1 Jan   1 Feb   1 Mar   1 Apr   1 May (2014)

-0.6   -0.4   -0.2   0   0.2   0.4   0.6   Velocity, cm/s

10/31 - 11/06, new moon

11/14 - 11/20, full moon

Depth, m

Time of day, h

Velocity, cm/s
-0.6   -0.4   -0.2   0   0.2   0.4   0.6

Petrusevich, V., Dmitrenko, I. A., Kirillov, S. A., Rysgaard, S., Falk-Petersen, S., Barber, D. G., Boone, W. and Ehn, J. K.: Wintertime water dynamics and moonlight disruption of the acoustic backscatter diurnal signal in an ice-covered Northeast Greenland fjord, J. Geophys. Res. Ocean., 121(7), 4804–4818, doi:10.1002/2016JC011703, 2016.

We added the following lines:

**Lines 344-347:** In certain cases vertical velocity actograms can be used for estimating swimming direction and velocity (Petrusevich et al., 2016) when for estimation of swimming direction actograms are averaged for layers of several meters depth and for velocity estimation individual profiles were averaged over a period of few days. This method works well when there is no tidal signal to be subtracted from the vertical velocity data, otherwise, it makes computation rather complicated

In our current study, we chose to focus primarily on analyzing vertical velocity wavelets to get power spectrum for the semidiurnal tidal currents and their spring-neap and seasonal variability. We then compared them to VBS maximums, doing statistical correlation analysis (Figure 5)

*Can you back any of your claims with numerics? Yes, you can see a pattern but is it statistically significant? E.g. calculate mean backscatter in daylight hours vs night hours*

For reviewers' reference we generated a graph (below) of difference in VBS between noon and midnight (not included in the revised manuscript).

[Figure]

We have run F-statistics to find statistically significant difference and added to the revised manuscript:

**Lines 175-177:** The mean difference between noon-time and midnight VBS was ~9±1dB at 96-100m depth layer and -3dB±1 at 10-28 m layer. Running F-statistic test returned statistical significance with 95% confidence for VBS difference below 58 m and above 48 m.

*Lines 185-187: This description belongs in the methodology.*

We placed it here because we are doing post-processing of the actograms to remove the tidal signal.

*Lines 187-190: So what is the diurnal variation? What is the pattern you see? How does this match up with your results in Figure 2? There you seem to find the strongest signals in the deeper layers?*

There is a stronger tidal signal in the deeper layers, for vertical velocity actograms cleared from the tidal signal we can not claim that there is a stronger diurnal signal, that associated with DVM in the deeper layers, while seasonal V-shape is distinguishable.

*Lines 191 to 192: It is not clear to me what you have done here. Are you looking at semi-diurnal horizontal tidal currents? Or are you looking at horizontal tidal currents from the ADCP which are semi-diurnally dominated? How have you post-processed your current data? Have you removed any tidal signals? What bandwidth are you using? It looks like it is greater than the semi-diurnal frequency?*

Here we are looking at the ADCP vertical current record. We postprocessed (as described in the lines 203-207) ADCP vertical current signal to be used only in the actograms.

We have revised this section to make it more clear:

**Lines 206-212:** The change in vertical speed associated with spring tide is present on the vertical velocity actograms in a form of slanted strips of 14-day periodicity, with amplitude increasing with depth and reaching maximum values in the range of 10-15 mm/s. The vertical velocity actograms were post-processed (Figure 4f-i) to remove the semi-diurnal tidal components ($M_2$ and $S_2$) from the vertical velocity data which otherwise would create tidal background signal in a form of slanted strips of 14-day periodicity on the actograms (Figure 4a-d). A tidal harmonic analysis was performed for the vertical velocity time series using T_Tide toolbox for Matlab (Pawlowicz et al., 2002).

*Line 200-204: This belongs in the methodology. What do these parameters mean for your data?*

We placed this in the results because it is a result of signal post-processing and it will be addressed in discussion section. These parameters are specific for application of more general Morse Wavelet transform for analysis of oceanic currents (instead of a standard Morlet wavelet), this is addressed in the papers we cite (Lilly, 2017, 2019; Lilly and Gascard, 2006; Lilly and Olhede, 2009).

*Line 200-201: I don't see a semi-diurnal signal in the currents – this is a spring-neap signal*

In the current actograms there is slanted signal which is actually – semidiurnal signal. Spring neap would not be seen in this case, just the periodicity of the slants on the actogram points to the periodicity of spring – neap signal. This is because it is just 'amplitude modulation' type of signal, that's why we used wavelet and we can see spring-neap signal in wavelet graphs.

*Lines 194-195: Yes, horizontal tidal currents tend to be at least an order of magnitude larger than vertical ones.*

We revised this section as:

**Lines 219-220:** The power spectrum range for horizontal velocities was, in general, over one order higher than for vertical velocity, which is consistent with the fact that horizontal tidal currents tend to be at least an order of magnitude larger than vertical ones.

*Section 4: It is not quite clear to me what the point of this exercise is? You show that you have stronger horizontal and vertical currents during spring tides and the opposite for neap tides. This is commonly known. So where does the link to DVM come in? Where do you show the tidal modulation your title promises? And you have removed the 12 hr signal that would give you the DVM? In Hudson Bay, the tides vary throughout the year due to changes in ice cover and stratification – how do you separate these effects from DVM?*

Here we compare VBS at 92 m with vertical velocity wavelet (which was calculated for actual non-postprocessed vertical velocities without removal of tidal signal). We removed tidal signal only from the second set of vertical velocity actograms. Wavelets allows us to separate semi-diurnal signal from diurnal. An example of similar use of wavelets is presented in our previous paper Petrusevich et. al. 2016.

*Figure 5. Describe the results from panel 5d in the text. How do you calculate your correlation coefficient? What time window is used? How did you obtain your backscatter time series? You obtain both significant positive and negative correlations, why are the negative ones not shaded pink?*

We did not shadow pink the negative correlations because negative correlations are artificial and have no physical meaning. We used 14 days window (as specified in the text).

We revised this portion of the manuscript accordingly:

**Line 365-369:** For 92 m depth, the 14-day running correlation (Figure 5d, green line) between midnight VBS (blue line) and vertical velocity wavelet (red line) was calculated. Correlations exceeding ±0.53 are statistically significant at the 95% confidence level (Figure 5d, yellow shading). Pink shading identifies the events when this statistically significant positive correlation was observed. Negative correlations are artificial and have no physical meaning.

*Line 214: Provided you know you are looking at zooplankton – see general point 1*

We revised the discussion section accordingly:

**Lines 241-250:** Even though acoustic backscatter from the single-frequency ADCP does not provide any information on the identity of zooplankton species involved in DVM but signal strength can provide an indication of zooplankton presence provided there is information on the zooplankton species. Sound is effectively scattered by objects of the size of the wavelength. For 300 kHz ADCP, it is about 5 mm. It is known that zooplankton species with body size less than the wavelength by an order of magnitude (in our case 0.5-5mm) are capable of creating strong backscatter when there is a sufficient abundance of them in the water column (Cisewski and Strass, 2016; Pinot and Jansá, 2001). The backscatter strength of zooplankton species also depends on their acoustic properties, such as shape, internal structure, orientation in the water column and body composition, that causes a difference between the speed of sound in their bodies and surrounding seawater (Stanton et al., 1994, 1998a, 1998b).
.

*Line 217: How do you know it's 5 mm – what is the size range on either size? What uncertainty exists here? What about objects larger than 5 mm? Or sediment?*

While it is true that 300kHz ADCP can be used for suspended sediment transport monitoring (Venditti et al., 2016), there are some general considerations that need to be taken into account. 300 kHz ADCP were used mostly in the rivers with high sediment loads (hundreds of mg/L).  The mooring was located  ~190 km north-east from the the Churchill river which does not create a significant plume of sediments into the system. Our mooring turbidity sensor located at 41m depth did not recorded values higher than 34 FTU which corresponds with TSS of ~30 mg/L, with average 7 FTU or TSS about ~5 mg/L. At 100 m depth we do not expect high levels of sediments from resuspension. Another consideration is the fact that sound is effectively scattered by objects of the size of the wavelength. For 300 kHz ADCP, it is about 5 mm (lines 235-241). Thus, smaller scatterers, like sediments, phytoplankton, etc. can be effectively eliminated as potential scatterers. This allows us to consider zooplankton as main scatters in our case.

There is a number of papers by Stanton et al. (1994, 1998a, 1998b) addressing the target strength of different zooplankton species for different frequencies. Sediments can also detected by ADCP VBS but it will be a background signal and will not show any patterns that are typical for DVM.

For better particle size discrimination, it is recommended to use dual frequency ADCP. ADCP has their limitation due to their beam configuration and without going too deep into engineering side of this question, it is established that mean particle size that is distinguished by 300kHz ADCP is 0.5 to 5 mm (Jourdin et al., 2014).

Jourdin, F., Tessier, C., Le Hir, P., Verney, R., Lunven, M., Loyer, S., Lusven, A., Filipot, J.-F. F., Lepesqueur, J., Hir, P. Le, Verney, R., Lunven, M., Loyer, S., Lusven, A., Filipot, J.-F. F. and Lepesqueur, J.: Dual-frequency ADCPs measuring turbidity, Geo-Marine Lett., doi:10.1007/s00367-014-0366-2, 2014.Stanton, T. K., Wiebe, P. H., Chu, D., Benfield, M. C., Scanlon, L., Martin, L. and Eastwood, R. L.: On acoustic estimates of zooplankton biomass, ICES J. Mar. Sci. J. du Cons., 51(4), 505–512, doi:10.1006/jmsc.1994.1051, 1994.

Stanton, T. K., Chu, D. Z., Wiebe, P. H., Martin, L. V and Eastwood, R. L.: Sound scattering by several zooplankton groups. I. Experimental determination of dominant scattering mechanisms, J. Acoust. Soc. Am., 103(1), 225–235, doi:10.1121/1.421469, 1998a.
Stanton, T. K., Chu, D. Z. and Wiebe, P. H.: Sound scattering by several zooplankton groups. II. Scattering models, J. Acoust. Soc. Am., 103(1), 236–253, doi:10.1121/1.421110, 1998b.

We have added the following lines to the revised manuscript:

**Lines 252-271:** It should be mentioned that 300kHz ADCP can be effectively used for suspended sediment transport monitoring (Venditti et al., 2016), but here are some general considerations that need to be taken into account. 300 kHz ADCPs were used for suspended sediment monitoring mostly in the rivers with high sediment loads (hundreds of mg L-1).  Our mooring was located  ~190 km north-east from the Churchill river which does not create a significant plume of sediments into the system. The mooring turbidity sensor located at 41m depth did not record values higher than 34 FTU which corresponds with TSS of ~30 mg/L, with average turbidity of 7 FTU which corresponds with TSS ~5 mg/L. At 100 m depth, we do not expect high levels of sediments from resuspension. Also taking into consideration the fact that that sound is effectively scattered by objects of the size of the wavelength and that the mean particle size detected by 300 kHz ADCP is in the range of 0.5 to 5 mm (Jourdin et al., 2014), sporadic smaller scatterers, like sediments, phytoplankton, etc. can be effectively eliminated as potential scatterers. This allows us to consider zooplankton as the main scatterers in our case.
Fish also can be detected with the ADCP used. It should be noted though that large mesopelagic fishes are rare in the Canadian Arctic (Berge et al., 2015a). Arctic cod (Boreogadus saida) is the dominant pelagic fish in the Canadian Arctic (e.g. Benoit et al., 2008; LeBlanc et al., 2019) therefore the acoustic signals related to fish are generally assumed to be only Arctic cod. The distribution of Arctic cod is known for regions such as the Beaufort Sea (Geoffroy et al., 2016) and Baffin Bay (LeBlanc et al., 2019). However, there is little known for Hudson Bay. It is expected that Hudson Bay Arctic cod behave similarly, with adult aggregations near the bottom in deep waters and young (year 1/2) and larval stages in surface aggregations. The young cod are ice-associated during the winter period, i.e., no migration to depth. As such, any backscatter associated with near-surface young cod would have been removed as part of the removal of the top 8 m of backscatter during post-processing.  Arctic Cod do not school. So, its presence in the proximity of the mooring will be more sporadic and acoustic backscatter will be significantly less than the backscatter from much more abundant zooplankton.

*Line 224: You are assuming that your trap is reflective of what's in the water column. Did you check this or can you prove this is correct any other way?*

We added a paragraph in Discussion 5.1

**Lines 282-287:** The zooplankton caught in our sediment trap provide general information on the zooplankton community composition and its change over the course of the year near the mooring location. Sediment trap samples may not quantitatively reflect zooplankton composition in the water column due to species-specific collection efficiencies. Comparisons between net and trap samples from Franklin Bay

indicate that the abundance of L. helicina and some species of copepods could be estimated from sediment traps whereas the abundance of other key species, such as C. hyperboreus, could not be accurately estimated from sediment trap samples (Makabe et al., 2016).

Also, the zooplankton assemblages in the trap samples were compared to net sampling conducted by Estrada et al (2012) in Hudson Bay. The dominant species of the net sampling survey are present in the sediment trap with a similar relative abundance of the dominant taxa. **(Lines 93-96, 273-281)**.

*Line 262-237: This needs to be backed up with an appropriate description of your results*

This general statement is addressed in the same paragraph by comparing the sediment trap catch with VBS at different depths.

We would like to thank Anonymous Reviewer #1 for all these helpful comments.

Regards,

On behalf of all authors

Vladislav Petrusevich

---

## Author Comment (AC2) · 30 Jan 2020

**Response to review comments to 'Impact of tidal dynamics on diel vertical migration of zooplankton in Hudson Bay' from Anonymous Referee # 2**

We highly appreciate helpful comments and suggestions from Anonymous Referee #2. In the following, *the comments by the reviewer are in italics* and our responses to the comments are in normal characters. The revised manuscript text is underlined. **The line numbering (in bold)** is referenced to the marked-up manuscript version.

**Review comments to 'Impact of tidal dynamics on diel vertical migration of zooplankton in Hudson Bay' by Petrusevich et al. Anonymous Referee # 2**

*The paper is very well written and with a clear and concise message. I have a few comments / questions:*

*1) line 55, objective 3: Why is not solar light mentioned here?*
It was our omission, thank you for pointing it out. We have added solar light as well **(line 70)**.

*2) How do you separate actively migrating from passively sinking (dead) organisms?*

Passively sinking organisms will produce just a background VBS. While migrating organism will have a periodic pattern that is clearly seen in VBS, especially in VBS actograms. Also taking into account life span of the zooplankton species in Hudson Bay, there is very unlike that mortality rate will be comparable to the number of the individuals actively participating in daily DVM cycle.

*3) Line 125. Ice thickness measured by ADCP - does there exist any groundtruthing data for this method? There are no references provided, except one that does not seem to be relevant? This needs to be updated / clarified*

I have added 4 references to the previous works that used ADCP for ice draft measurements: (Banks et al., 2006; Björk et al., 2008; Shcherbina et al., 2005; Visbeck and Fischer, 1995).

*4) There is a basic understanding or basis for a DVM pattern regulated by light that is not really presented, but which is essential to the entire manuscript. I would strongly suggest that the authors first describe this general and consistent DVM, and then focus on how this is disrupted. One way of doing this would be to compared noon with midnight mean position in the water column throughout the entire data series.*

Mean position method is working well when is used in actual zooplankton sampling: for example the paper (Munk et al., 2015). They used the following formula:

$$CM = \frac{\sum a_j \times b_j}{\sum b_j}, j = 1, \dots, n$$

Where $a_j$ is the mean depth of sampling interval j, and $b_j$ is abundance within sampling intervals, j. There is a different mean position for various species and their variation for day and night also varies for various species. Below is a figure from Munk et al., 2015:

[Figure]

From this graph one can see that for some types of Calanus the difference between day and night is not really big just around 1-2m.

In our case we are not doing the actual sampling but just analyzing the backscatter created by composition of various migrating species.

I did a quick estimate for mean position based on VBS for my dataset in Matlab:

[Figure]

The only thing we can say that this method gives very close results for both day and night and that depth is 51-52m, which is typical for juvenile Calanus glacialis and Pseudocalanus spp.

Munk, P., Nielsen, T. G. and Hansen, B. W.: Horizontal and vertical dynamics of zooplankton and larval fish communities during mid-summer in Disko Bay, West Greenland, J. Plankton Res., 37(3), 554–570, doi:10.1093/plankt/fbv034, 2015.

*When these issues are sorted, I recommend that the manuscript is accepted for publication*

We would like to thank Anonymous Reviewer #2 for all these helpful comments.

Regards,

On behalf of all authors

Vladislav Petrusevich

---

## Author Comment (AC3) · 30 Jan 2020

**Response to review comments to 'Impact of tidal dynamics on diel vertical migration of zooplankton in Hudson Bay' from Anonymous Referee # 3**

We highly appreciate helpful comments and suggestions from Anonymous Referee #3. In the following, *the comments by the reviewer are in italics* and our responses to the comments are in normal characters. The revised manuscript text is underlined. **The line numbering (in bold)** is referenced to the marked-up manuscript version.

*General*
*This work exploits acoustic data from an ADCP moored over an annual cycle, backed with zooplankton identification in sediment trap samples, to document the seasonal dynamics of zooplankton DVM at a seasonally ice-covered site in Hudson Bay. The data analyses sound complete but the highlighted results do not seem particularly novel in the way they are presented. Maybe one approach to deal with this perception is to work on a more thorough comparison of the patterns observed in Hudson Bay with other regions.*

Hudson Bay is a very interesting and unique seasonally ice-covered region that was in general not sufficiently studied. Being one of the largest inland seas, there were very few studies on zooplankton composition and behaviour (Estrada et al., 2012; Runge and Ingram, 1991). While the phytoplankton and zooplankton assemblages in Hudson Bay resemble those in the Arctic Ocean, another unique feature that we discuss in our paper it is interaction of zooplankton and particularly DVM with the strong tides that are characteristic of Hudson Bay.

*More in-depth interpretation of the linkages between the acoustic observations and zooplankton biology would also help this work. In particular, tidal effects on DVM seem to be emphasized by the title but this does not appear that well in the Discussion.*

Thank you for pointing out the importance on the emphasis on tidal dynamics on DVM. In this paper we tried to analyze the zooplankton response to spring tide from the acoustic VBS data. The possible interpretation was that the barotropic tide interacts with bottom topography generating tidal flow diverging and converging vertically. It seems that zooplankton tends to avoid spending additional energy swimming against the vertical flow. This response of zooplankton is consistent with the zooplankton tendency to stay away from the layers with enhanced water dynamics and to adjust its DVM accordingly, which we previously observed in Young Sound fjord in NE Greenland when there was polynya induced circulation in the fjord.
We attempted to cover the tidal background in Hudson Bay in introduction **(lines 73-75),** then in results we presented the tidal effect on VBS signal **(lines 178-179)**, vertical velocity actograms **(lines 201-205)**, wavelet analysis **(201-221)**, discussion **(lines 335-358)** and conclusion **(lines 373-377).**

*The structure of the manuscript needs to be better strengthened as there are pieces of different sections that should belong to other ones, as detailed in the specific comments. The title takes into account only one aspect addressed by this work. There is also an important issue that should be addressed either in the Introduction of*

*the Discussion: is the trap a valid way to identify the scatterers?*

Sediment traps were successfully used for qualitative assessment of the species responsible for acoustic backscatter (Berge et al., 2009; Cottier et al., 2006; Ota et al., 2008; Wallace et al., 2010; Willis et al., 2006, 2008). It is an accepted and recommended practice to simultaneously use samples collected by sediment traps to better under acoustic instruments backscatter signals (Makabe et al., 2016).

The sediment trap captures sinking zooplankton. There is a low probability that as the zooplankton sink they would reverse and search for a way out.  When compare to literature on zooplankton in Hudson Bay (see Estrada et al. 2012), the species collected within this sediment trap are representative of the zooplankton community we would expect to find in this system. Gelatinous zooplankton recorded by Estrada et al. 2012 also occurred in the zooplankton samples, however, at low numbers (data not shown). It is possible that gelatinous species are underestimated due to issues associated with preservation. This is noted at **lines 244-251, 273-287** in the paper.

We revised the introduction of the Discussion adding the following lines:

**Lines 282-287.** The zooplankton caught in our sediment trap provide general information on the zooplankton community composition and its change over the course of the year near the mooring location. Sediment trap samples may not quantitatively reflect zooplankton composition in the water column due to species-specific collection efficiencies. Comparisons between net and trap samples from Franklin Bay indicate that the abundance of *L. helicina* and some species of copepods could be estimated from sediment traps whereas the abundance of other key species, such as *C. hyperboreus*, could not be accurately estimated from sediment trap samples (Makabe et al., 2016).

*Specific comments*
*Title*
*The title does not reflect the scope of this work properly since the tidal effects was only*
*one part of the Discussion*

Besides mentioned portion of the Discussion, we covered the tidal background in Hudson Bay in introduction **(lines 76-78),** then in results we presented the tidal effect on VBS signal **(lines 183-184)**, vertical velocity actograms **(lines 206-212)**, wavelet analysis **(215-228)** and conclusion **(lines 402-406).**

*Abstract*
*Line 13-14: Give the information on potential migrators instead of telling that they could*
*be identified.*

Thank you for this suggestion. We have added the following lines to the abstract:

**Lines 13-15:** The sediment trap collected five zooplankton taxa including two calanoid copepods (*Calanus glacialis* and *Pseudocalanus* spp.), a pelagic sea snail (*Limacina helicina*), a gelatinous arrow worm (*Parasagitta elegans*) and an amphipod (*Themisto libellula*).

*Line 14: "migrating scatters"? what does that mean? How can a scatter migrate?*
Corrected. Thank you for pointing this typo. We have changed to migrating scatterers.

*Introduction*

*Line 20: I would remove "synchronized" from the sentence, as DVM doesn't have to be synchronized to transport C and N to depth. Furthermore, "synchronized" is used in the following sentence that explains DVM.*

Thank you for pointing it out. We have revised the beginning of introduction:

**Lines 21-22:** The diel vertical migration (DVM) of zooplankton is a synchronized movement of individuals through the water column and is considered to be the largest daily synchronized migration of biomass in the ocean (Brierley, 2014).

*Line 28: Explain better why this question needs to be addressed*

We added the lines in introduction on general DVM patterns (lines 30-35).

**Lines 30-35:** There are three general DVM patterns: (1) The most common one is *nocturnal* when zooplankton ascends around sunset and remains at upper depth during the night, around sunrise descending and remaining at depth during the day (Cisewski et al., 2010; Cohen and Forward, 2002). (2) Then there is a *reverse* pattern when zooplankton is ascending up at dawn and descending at dusk (Heywood, 1996; Pascual et al., 2017). And finally, (3) there is *twilight* DVM pattern when zooplankton is ascending at sunset, then descending around midnight, then again ascending and finally descending at sunset (Cohen and Forward, 2005; Valle-Levinson et al., 2014). This pattern sometimes is called *midnight sink*.

*Line 39: remove "to" after "help"*
Corrected **(line 49)**

*M&M*
*Line 79: It is "Macrozooplankton" we are talking about here and not "Microzooplankton"*
Thank you for pointing out that typo. Sure, we are not talking about microzooplankton. The samples collected in the sediment trap represent both mesozooplantkon (0.5-2mm for our samples) as we macrozooplankton (>2 mm, e.g. the *Parasagitta elegans*).

*Line 80: "Parasagitta" instead of "Sagitta"*
Thank you for pointing out that typo. Corrected

*Lines 83-85: This information does not fit in here in the description of the study area.*
*The authors should find a more proper place to use it if needed. The whole paragraph*
*on zooplankton should be moved somewhere else.*

We just follow similar introduction style like used in Estrada et al., 2012 and Harvey et al., 2001, where after physical description of the sampling site there is given info on zooplankton community composition in Hudson Bay.

Estrada, R., Harvey, M., Gosselin, M., Starr, M., Galbraith, P. S. and Straneo, F.: Late-summer zooplankton community structure, abundance, and distribution in the Hudson Bay system (Canada) and their relationships with environmental conditions, 2003–2006, Prog. Oceanogr., 101(1), 121–145, doi:https://doi.org/10.1016/j.pocean.2012.02.003, 2012.

Harvey, M., Therriault, J.-C. and Simard, N.: Hydrodynamic Control of Late Summer Species Composition and Abundance of Zooplankton in Hudson Bay and Hudson Strait (Canada), J. Plankton Res., 23(5), 481–496, doi:10.1093/plankt/23.5.481, 2001.

*Line 92: the sampling area of this trap is very small and may cause a bias in zooplankton catching toward the smaller individuals that need to be addressed.*

This is a valid concern, but we should mention, that regardless of its compact size the trap caught a wide range sizes, including large shrimp and long zooplankton. Sediment traps are designed to capture the sinking flux of material from an area above the trap. Based on previous assessments of zooplankton in Hudson Bay, larger expected species (e.g., *Parasagitta elegans*) were present in the trap.

*Line 137: a citation is needed to back the information on the size fraction effectively sampled by the ADCP*

Thank you for this note. Corrected, added (Cisewski and Strass, 2016; Pinot and Jansá, 2001)

*- "Motoda" instead of "Motodo"*
Thank you for pointing out that typo. Corrected

*Results Line 143: Does that mean that in a matter of a few days, the ice thickness reached 0.4 m?*

Yes, that's what happening normally in HB. Figure 2 shows that it took about a week for the ice to reach that thickness. Oceanic heat in the Bay delays ice growth, and from the onset of ice formation air temperatures are very low.

*Line 145: remove "the" Line 146: replace "were" by "are"*

Thank you for pointing these typos, we got them corrected.

*Line 154: replace "scatters" by "scatterers", here and elsewhere.*

Thank you for pointing it out, we have corrected in the manuscript.

*This sentence is a piece of the Methods and would fit better in the previous section.*

We would agree that it might be more appropriate in the methods, but in methods we described a general idea of calculation of volume backscatter strength, and here we are presenting a specifically applied results (VBS at noon, midnight and actograms).

*Line 156: the part on DVM in this sentence is interpretation of Results and would fit better into the Discussion.*

Here we just briefly point out the difference between noon and midnight VBS and expand it more in discussion.

*Line 158: statistics?*

For reviewers' reference we created a graph (below) of difference in VBS between noon and midnight (not included in the revised manuscript).

[Figure]

We have run F-statistics to find statistically significant difference and added to the revised manuscript:

**Lines 175-177:** The mean difference between noon-time and midnight VBS was ~9±1dB at 96-100m depth layer and -3dB±1 at 10-28 m layer. Running F-statistic test returned statistical significance with 95% confidence for VBS difference below 58 m and above 48 m.

*Line 160: "midnight bottom scatters layer" by "layer of midnight bottom scatterers"*
Corrected

*Line 162: "maxima" instead of "maximums"*
Corrected

*Line 164: remove "observed"*
Corrected

*Line 166: "shape shows a similar overall shape..." too many "shape" and "overall" here*
Changed this passage into:
**Lines 187-190:** Overall, VBS actograms show a similar shape to that of the under-ice solar illumination actogram (Figure 3i). This resemblance in shape is outlined by reduced VBS at 8 and 20 m actograms (Figure 3d-e) and enhanced at 60, 80 and 92 m actograms (Figure 3f-h) during dawn and dusk.

*Line 205: remove brackets*
Corrected

*Line 208: It is "libellula", not "libellua"*
Corrected

*Line 208-209: This sentence does not provide results information.*

We have moved this section to the discussion **(Lines 272-273).**

*Discussion*
*The first paragraph of a discussion should give justice to the Results and novel knowl-*
*edge provided by the work and entice the reader to learn more about the issue. I would*
*turn the first sentence differently so that it would not look so much like it emphasizes*

*the weakness of the ADCP-based method to study zooplankton patterns.*

Thank you for this good suggestion. I have added intro sentences to this section as follows:

**Lines 238-243:** The presence of seasonal ice cover acts as a barrier to using traditional zooplankton sampling techniques. But using both moored or ice-tethered ADCPs in high latitudes had been successful for studying zooplankton presence, behaviour and particularly DVM patterns (Darnis et al., 2017; Hobbs et al., 2018; Petrusevich et al., 2016; Wallace et al., 2010). Even though acoustic backscatter from the single-frequency ADCP does not provide any information on the identity of zooplankton species involved in DVM but signal strength can provide an indication of zooplankton presence provided there is information on the zooplankton species.

*Line 215: Studies like the one by Makabe et al (2016) address the issue of the useful-ness of sediment trap samples for the description of zooplankton community compo-sition and seasonal change by comparing zooplankton caught in sediment traps with ones sampled by plankton nets. What is found in the trap samples does not neces-sarily give a good picture of the zooplankton composition in the water column. The trap might miss the importance of scatterers that are not well sampled by the small-aperture trap. Themisto might be quite under sampled by the small trap. Furthermore, traps are known to oversample pteropods that stop swimming and sink when they touch the mooring line. Some change in behavior influencing the depth range of zooplankton will also have an impact on trap catching efficiency. This has to be kept in mind and mentioned.*
*Makabe, R., Hattori, H., Sampei, M., Darnis, G., Fortier, L., Sasaki, H., 2016. Can sediment trap-collected zooplankton be used for ecological studies? Polar Biol., doi:10.1007/s00300-00016-01900-00307.*

Thank you for pointing this out I have restructured and added the following paragraph:

**Lines 282-287:** The zooplankton caught in our sediment trap provide general information on the zooplankton community composition and its change over the course of the year near the mooring location. Sediment trap samples may not quantitatively reflect zooplankton composition in the water column due to species-specific collection efficiencies. Comparisons between net and trap samples from Franklin Bay indicate that the abundance of *L. helicina* and some species of copepods could be estimated from sediment traps whereas the abundance of other key species, such as *C. hyperboreus*, could not be accurately estimated from sediment trap samples (Makabe et al., 2016).

*Line 235: DVM patterns have already been documented in another part of Hudson Bay (Runge and Ingram 1991). The authors should give credit to the pioneer study in this paragraph.*
*Runge, J.A., Ingram, R.G., 1991. Under-ice feeding and diel migration by the planktonic copepods Calanus glacialis and Pseudocalanus minutus in relation to the ice algal production cycle in southeastern Hudson Bay, Canada. Mar. Biol. 108, 217-225.*
*C4*

Thank you for pointing out that good pioneering paper, but it was from a different part of Hudson Bay (coastal areas of South East HB), where there is a significant freshwater inflow, which is different from our mooring location, and they showed a different/additional reason for migration. So, I added this paper to the references in the intro **(line 83)** rather than using it in this section.

*Line 245: Well, we do not have the elements of information yet to tell if this pump is important or not. Importance would depend on the real scatterers, and depth and stratification state of the water column.*

We have removed the world important.

*Line 246: "DVM" and not "DMV"*
Corrected

*Line 247: "vertical transport of elements" instead of "vertical energy transfer"*
Corrected

*Line 248: I don't think that it is worth introducing the next sections in that way. Normally a logical suite of sub-sections should be enough.*
*Or replace by something like : "the acoustic data at hand are not valid to quantify zooplankton biomass involved in DVM. However, we can use them to document and understand better important aspects of DVM, such as: links between its seasonal cycle and dynamics of sea-ice cover and under-ice illuminance, and the effects of wind storms and tides on DVM patterns".*

Thank you for this suggestion and we reworded this section following your suggestions.

**Lines: 303-309:** Regardless, there is a pump of carbon/nitrogen occurring within Hudson Bay based on zooplankton DVM, and seasonal differences (discussed in the next section) could impact this vertical transport of elements. The collected acoustic data at hand are not valid to quantify zooplankton biomass involved in DVM. However, we can use them to document and understand better important aspects of DVM, such as links between its seasonal cycle and dynamics of sea-ice cover and under-ice illuminance, and the effects of wind storms and tides on DVM patterns.

*Line 252: "south" instead of "southern location". In any case, this sentence should be rewritten to improve its clarity. Make the message straighter.*

Thank you for your suggestion, we have revised this passage as:

**Lines 311-314:** The mooring site is located 6° south of the Arctic circle and polar twilight zone. Hudson Bay located more south than other seasonally sea-ice covered Arctic and sub-Arctic regions where DVM was observed. In those locations, DVM during the winter was primarily controlled by twilight and the lunar light (Last et al., 2016; Petrusevich et al., 2016).

*In general, there are too many figure citations in this section. If the Results section is clearly written, there is no need to cite those figures again. The Discussion should take on from the Results described in the previous section.*

We have added to this section dusk and dawn VBS discussion **(lines 317-320)**, so it would be convenient to keep the citations to the figures.

**Lines 317-320:** During dawn and dusk there was reduced VBS at 8 and 20 m actograms (Figure 3d-e) and enhanced at 60, 80 and 92 m actograms (Figure 3f-h). These dawn and dusk absences and enhancements can be interpreted as an indication of zooplankton swimming behaviour during these periods, following nocturnal DVM pattern. The increased backscatter at dawn and dusk at 60 and 80 m actograms was observed regardless the presence of ice cover.

*Line 271: by definition, the trap does not measure abundance but a rate of capture or sinking in the case of inert particles. Thus, I fear that it can be too misleading to use the term "abundance" in that case even though it is mentioned that it is the abundance in the trap sample after 35 days of opening. This is because the rate will not necessarily be related to the abundance of organisms in the water column. This is a tricky issue that should be addressed carefully.*

We agree that the abundance in the trap should not be understood as abundance in the water column. But it is accepted term being used for sediment trap catches, for example the paper *Makabe et al (2016)* paper you recommended is using term abundance both for net catches and for sediment trap catches. Also Schröter et al., 2019 is using term 'abundance' when dealing with sediment trap catches.

Schröter, F., Havermans, C., Kraft, A., Knüppel, N., Beszczynska-Möller, A., Bauerfeind, E. and Nöthig, E.-M.: Pelagic Amphipods in the Eastern Fram Strait With Continuing Presence of Themisto compressa Based on Sediment Trap Time Series, Front. Mar. Sci., 6, 311, doi:10.3389/fmars.2019.00311, 2019.

*Line 280: one alternate explanation that should be discussed is that of different feeding patterns. Some non-visual predators like chaetognaths might not need to move that much if their zooplankton prey change their migration patterns as well etc..*

I have added the passage:

**Lines 360-363:** An alternative explanation of higher VBS at 8 m depth is a different feeding pattern for non-visual predators like chaetognaths (including P. elegans). While mature species are known to perform DVM, in some cases juvenile individuals were found near the surface during the daytime (Brodeur and Terazaki, 1999).

Brodeur, R. D. and Terazaki, M.: Springtime abundance of chaetognaths in the shelf region of the northern Gulf of Alaska, with observations on the vertical distribution and feeding of Sagitta elegans, Fish. Oceanogr., 8(2), 93–103, doi:10.1046/j.1365-2419.1999.00099.x, 1999.

*Line 281: Is it disruption of masking of the DVM signal? From the interpretation, it is not possible to understand if the storms act on the zooplankton responsible for the DVM patterns, or if other physical action produce backscatter that prevent the visualization of DVM. The paper should relate storms to zooplankton behavior or change the title of this sub-section, which then would much less relevant.*

This is a valid point, so we changed the title of that sub-section to: 5.3 Masking of DVM signal in the upper layer by storms **(line 348).**

*Line 288: remove "present"*

Corrected

*Line 291: "amount of" and not "amount in"; "provides" and not "provide"*

Thank you for pointing those typos. Got them corrected.

We would like to thank Anonymous Reviewer #3 for all these helpful comments.

Regards,

On behalf of all authors

Vladislav Petrusevich